# An hourly 0.02 ° total precipitable water dataset for all-weather conditions over the Tibetan Plateau through the fusion of observations of geostationary and multi-source microwave satellites

Qixiang Sun[1,2], Husi Letu[1, *], Yongqian Wang[3], Peng Zhang[4,5], Hong Liang[4], Chong Shi[1], Shuai Yin[1], and Jiancheng Shi[6], Dabin Ji[1, *]

[1]State Key Laboratory of Remote Sensing and Digital Earth, Aerospace Information Research Institute, Chinese Academy of Sciences, Beijing 100101, China
[2]University of Chinese Academy of Sciences, Beijing 100049, China
[3]College of Resources and Environment, Chengdu University of Information Technology, Chengdu 610225, China.
[4]Meteorological Observation Center, China Meteorological Administration, Beijing, 100081, China
[5]State Key Laboratory of Environment Characteristics and Effects for Near-space
[6]National Space Science Center, Chinese Academy of Sciences, Beijing 100190, China

*Correspondence to*: Dabin Ji (jidb@aircas.ac.cn); Husi Letu (husiletu@radi.ac.cn)

**Abstract.** The Tibetan Plateau (TP), known as the "Asian Water Tower", plays a critical role in the regulation of the water cycle in the region. Obtaining high spatiotemporal resolution, and all-weather total precipitable water (TPW) data is essential for understanding water vapor transport mechanisms, improving precipitation forecasting, and managing regional water resources over the TP. However, existing single-sensor remote sensing techniques cannot provide high spatiotemporal resolution TPW data under cloudy conditions. Multi-source fusion approaches often produce anomalous distributions in the fused TPW data due to inter-sensor biases, particularly over the complex terrain of the TP. This study proposed a multi-source remote sensing TPW fusion framework that integrates TPW products from eight microwave satellites and the Himawari-8/9 (H8/9) geostationary satellite to produce an all-weather TPW data with the highest spatiotemporal resolution at present. Methodologically, two correction strategies were developed. First, a bias correction approach was proposed using H8/9 TPW data as a reference to calibrate multi-source microwave remote sensing TPW and reduce inter-sensor discrepancies. Second, an adaptive correction method was created to improve the accuracy and spatial continuity of the fused TPW data under cloudy conditions. Based on the newly developed fusion framework, an all-weather TPW dataset with hourly temporal and 0.02° spatial resolution covering the TP from 2016 to 2022 was produced for the first time. The new dataset has been published by the National Tibetan Plateau Data Center and is available at: https://doi.org/10.11888/Atmos.tpdc.301518. Taking the 2017 product as an example, it was verified against GNSS TPW. The RMSE of the fused TPW product at the hourly scale was 3.79 mm, which was 10.82% and 6.19% lower than MIMIC-TPW2 and ERA5, respectively. Compared to ERA5 with a spatial resolution of 0.25°, the fused product achieves a 12.5-fold improvement in spatial resolution, which make it possible to significantly grasp the transportation of water vapor in the valley of Yarlung Zsangbo River. It also demonstrates higher reliability in station-sparse regions, providing high-quality, high-resolution vapor data to support vapor flux estimation and forecasting of extreme weather events over the TP.

# 1 Introduction

The Tibetan Plateau (TP), known as the "Asian Water Tower", plays a crucial role in the reception, storage, and redistribution of water vapor across Asia (Dong et al., 2024; Xu et al., 2008; Yao et al., 2012). Atmospheric water vapor over the TP shapes the spatial distribution of precipitation and surface runoff across both the TP and downstream regions (Shen et al., 2021; Chen and Yao, 2023). The resulting changes in hydrological processes provide essential support for downstream water resource management, ecosystem stability, and agricultural irrigation scheduling (Immerzeel et al., 2010; Yang et al.,

2012). Under cloudy conditions, interactions between water vapor and cloud microphysical processes further modify the surface shortwave radiation budget (Letu et al., 2023). However, the combined influence of complex terrain and strong local circulations over the TP results in pronounced spatiotemporal heterogeneity in water vapor distribution, often manifested as frequent small-scale vapor convergence and convective initiation. Capturing these fine-scale processes requires high-resolution water vapor data. Such data enhance the estimation of vapor fluxes and support quantitative analysis of the

interaction between humidity and precipitation (Zhang et al., 2024; Feng and Zhou, 2012; Trenberth and Guillemot, 1998). They also help identify localized vapor accumulation zones and convective triggers, ultimately improving the accuracy of hydrological assessments and forecasts of severe weather events (Chen and Yao, 2023).

Currently, studies of the spatiotemporal variation of water vapor over the TP rely on three main types of total water vapor content data: in-situ observations, reanalysis products, and satellite remote sensing data. These data sources provide different

levels of spatial coverage, temporal resolution, and accuracy. In situ observations, such as the Integrated Global Radiosonde Archive (IGRA, Durre et al., 2006), SumiNet (Ware et al., 2000), and the Crustal Movement Observation Network of China (CMONOC, Shi et al., 2008), retrieve water vapor through radiosondes or the Global Navigation Satellite System (GNSS), with typical errors of less than 2 mm. However, these observational networks have limited spatial coverage, and stations are especially sparse over the TP, making it difficult to capture the spatial gradients of water vapor. Reanalysis datasets, such as

the European Centre for Medium-Range Weather Forecasts Reanalysis v5 (ERA5), provide temporally and spatially continuous water vapor data, but its spatial resolution is relatively coarse (about 31 km, Hersbach et al., 2020). Satellite retrieval of total column water vapor mainly relies on three techniques: near-infrared (NIR), thermal infrared (TIR), and microwave (MW). These methods differ based on the type of sensor and the spectral bands. Satellite sensors such as the Moderate Resolution Imaging Spectroradiometer (MODIS) or Medium Resolution Spectral Imager (MERSI), when using

their near-infrared channels, retrieve total column water vapor based on the differential absorption principle (Abbasi et al., 2020; Ma et al., 2022b; Wang et al., 2021; Xu et al., 2022), achieving a spatial resolution of 1 km with an accuracy of 5–10% (Gao & Kaufman, 2003). This type of retrieval can only be applied under clear-sky conditions and is further limited to daytime due to its reliance on solar radiation. TIR-based retrievals of total column water vapor typically employ split-window algorithms, which utilize thermal infrared channels such as 7.0, 11.0, and 12.0 μm (Dalu, 1986; Guillory et al., 1993;

Labbi and Mokhnache, 2015; Liu et al., 2017). The TIR-based retrieval methods are applicable to both geostationary and polar-orbiting satellites, allowing observations during both day and night. However, TIR radiation cannot penetrate clouds,

so the TIR-based retrieval approaches are unable to retrieve water vapor beneath cloud cover. Microwave remote sensing operates at longer wavelengths, allowing it to penetrate clouds and retrieve water vapor under all-weather conditions. Satellite-based microwave sensors include microwave radiometers and microwave sounders. Over land, retrievals are
challenged by surface emissivity variability (Du et al., 2015). Ji et al. (2017) proposed an optimized retrieval algorithm for land surfaces based on AMSR-E brightness temperatures at 18.7 GHz and 23.8 GHz. The method introduces a new water vapor sensitive parameter derived from the polarization difference ratio, improves surface emissivity estimation, and incorporates digital elevation model (DEM) data to correct for terrain effects, thereby enhancing retrieval accuracy under complex surface conditions. In recent years, machine learning methods have been widely applied to the retrieval of
atmospheric parameters, offering both high speed and accuracy (Shi et al., 2021; Tang et al., 2025). For example, over land, microwave radiometer-based retrievals of water vapor using machine learning have achieved errors ranging from 2.70 to 3.84 mm (Bonafoni et al., 2011; Gao et al., 2022; Jiang. N et al., 2022a; Xia et al., 2023). Microwave sounders, which typically include water vapor sensitive channels near 22.235 GHz and 183 GHz, can also retrieve water vapor using algorithms such as one-dimensional variational (1DVAR) methods (Liu & Weng, 2010). Over land, these retrievals yield
errors of approximately 5.03 to 5.94 mm (Boukabara et al., 2010). However, the coarse spatial resolution of these sounders (e.g., approximately 17 km for DMSP-F17) limits their effectiveness for high-resolution water vapor applications over the TP.

Although various atmospheric water vapor data sources have their own advantages, a single data source is still difficult to meet the demand for high-resolution, all-weather total atmospheric water vapor content inversion in the TP region. Among
them, different types of remote sensing observations have complementary characteristics. For example, infrared remote sensing provides high spatial resolution under clear skies, while microwave remote sensing enables all-weather retrieval of atmospheric parameters. Therefore, by developing multi-source remote sensing observations fusion methods and complementing the advantages of multiple data sources, it is possible to achieve all-weather high-spatiotemporal-resolution total atmospheric water vapor content (also known as total precipitable vapor, hereinafter referred to as TPW) data
reconstruction.

In the pursuit of obtaining all-weather, high-resolution TPW data, improving spatial resolution and filling observational gaps under cloudy conditions in complex terrain continue to be pivotal challenges in TPW data fusion studies. In recent years, researchers have mainly adopted methods such as spatial interpolation, spatiotemporal fusion, and machine learning to enhance the spatial resolution of TPW data and supplement the missing data under clouds (Li & Long, 2020; Zhao et al.,
2022; Zhang. B et al., 2019). The basic idea of spatial interpolation methods is to utilize the spatial distribution characteristics of existing observation points and infer the TPW data in unobserved areas based on the spatial correlation or distance weights between adjacent points. Common techniques include inverse distance weighting (IDW), Kriging interpolation, and Spherical Cap Harmonic Analysis (SCH) (Li, 2004; Alshawaf et al., 2015; Zhang. B et al., 2019), which have been widely applied in the fusion of GNSS, InSAR, and reanalysis data to reconstruct high-accuracy, all-weather TPW
datasets. However, these interpolation approaches also face limitations: (1) When large-scale cloud systems cover most of

the observation areas and station observations are sparse, the reliability of interpolation methods will decrease, which may lead to boundary errors and spatial smoothing effects; (2) Spatial interpolation methods are mainly based on statistical relationships and do not fully consider atmospheric motion such as water vapor transport and convective development. The core idea of spatiotemporal fusion methods is to match high spatial resolution but low temporal resolution polar-orbiting satellite data (such as MODIS TPW products) with high temporal resolution but low spatial resolution reanalysis data (such as ERA5) to obtain all-weather high-resolution water vapor data (Li & Long, 2020). However, during the monsoon season, cloud cover becomes particularly severe in regions such as the Hengduan Mountains in southeastern TP, with cloud coverage exceeding 70% (Bao et al., 2019; Bao et al., 2024). This significantly limits the availability of high spatial resolution optical remote sensing observations, thereby constraining the applicability of spatiotemporal fusion methods. Zhao et al. (2022) proposed a two-step TPW fusion method and tested it in Yunnan Province, China. The first step reconstructed high spatial resolution TPW data using polynomial fitting and spherical harmonic functions, in combination with ERA-Interim and the Global Pressure and Temperature 2 wet model (GPT2w). The second step integrated high temporal resolution GNSS observations to achieve hourly 0.1° resolution TPW data reconstruction. Compared to earlier studies (Li & Long, 2020), this method resolves the issue of the lack of continuous temporal observations in the high-resolution TPW data sequence during fusion. However, this approach depends on GNSS station observations, and the sparse distribution of such stations in complex terrain regions like the TP (Zhang. H et al., 2021) limits its applicability. In recent years, machine learning algorithms such as neural networks (NN), random forests (RF), and various gradient boosting tree (GBT) models have been introduced into TPW data fusion studies, giving rise to a range of alternative fusion strategies (Du et al., 2025; Lu et al., 2022; Ma et al., 2022a, 2022c; Ma et al., 2023; Sun et al., 2024; Xiong et al., 2021; Zhang. B et al., 2021). These methods integrate multi-source water vapor data, together with auxiliary variables such as elevation, vegetation index, time, latitude, longitude, and meteorological data to construct nonlinear mapping relationships for reconstructing all-weather, high-accuracy TPW data. However, most existing approaches rely on GNSS or reanalysis data as reference inputs. The former suffers from sparse station coverage over the TP, while the latter has coarse spatial resolution (> 30 km), making them insufficient for accurately characterizing high-resolution water vapor structures in complex terrain regions. In contrast, Sun et al. (2024) proposed a two-step fusion framework that does not rely on GNSS or reanalysis data as reference inputs: the first step derives all-weather, high-resolution microwave-based TPW data using newly developed water vapor sensitive indicators, and the second step improves the accuracy of microwave remote sensing retrievals under cloudy conditions by fusing high-precision NIR TPW observations. This method reduced TPW retrieval errors by 18.79% compared to single-sensor observations over China, including most of the TP, and successfully reconstructed high-resolution water vapor fields beneath clouds in regions with sparse GNSS coverage. In addition, Du et al. (2025) proposed a TPW fusion approach that integrates NIR, TIR, and MW observations, and applied it in Australia. This study employed an iterative tropospheric decomposition (ITD) technique to separate TPW into stratified and turbulent components for high-resolution reconstruction, achieving ultra-fine spatial resolution TPW estimates at 0.001°. However, like the limitations in Sun et al. (2024), the temporal resolution of the reconstruction results remains low, and it cannot provide hourly TPW data. In summary, although

the aforementioned methods have achieved notable advances in high-spatial-resolution TPW estimation, ensuring temporal continuity while maintaining spatial accuracy in topographically complex and data-sparse regions such as the TP remains a key challenge in current TPW fusion research.

In terms of enhancing the temporal continuity of TPW data, researchers mainly adopt extrapolation interpolation methods, using multi-source satellite observations or combining reanalysis data to fill in the gaps in observation time, achieving

higher-frequency water vapor monitoring. The core idea of the extrapolation and interpolation method that only uses multi-source microwave data is to utilize the TPW data from multiple microwave satellites and perform interpolation through statistical methods, motion compensation techniques, or extrapolation to fill gaps between satellite overpasses (Ermakov et al., 2016; Kidder and Jones, 2007). Kidder and Jones (2007) matched observations from different microwave satellites and applied a cumulative probability distribution function (CPDF) correction to reduce inter-sensor inconsistencies. This

approach optimized the temporal interpolation process of TPW and enabled the generation of all-weather TPW datasets at 0.25° resolution with 1–3 hour intervals. Ermakov et al. (2016) employed motion estimation and compensation techniques to address orbital gaps in merged multi-source microwave observations. They estimated TPW displacement trends using well-matched data blocks from adjacent scenes and extrapolated to fill missing regions, producing TPW datasets with 0.125° spatial resolution and 1.5 hour temporal intervals. The characteristic of such extrapolation filling methods lies in not relying

on numerical model data. Moreover, there are also methods that utilize reanalysis data (Wimmers and Velden, 2011; Sun et al., 2021; Ma et al. 2022a). Sun et al. (2021) proposed an optimal interpolation (OI) approach, using ERA-Interim data as the background field and incorporating TPW observations from multiple microwave satellites to generate a daily 0.25° TPW product with multi-year global ocean coverage. Additionally, Wimmers and Velden (2011) proposed an advection method that combines multi-source microwave observations and Global Forecast System (GFS) forecast wind field data to produce

the Morphed Integrated Microwave Imagery at CIMSS TPW product version 2 (MIMIC-TPW2), a global hourly 0.25° TPW product. The core assumption of these methods is that TPW is mainly transported by the horizontal wind field, and the displacement vectors calculated based on wind field data are used to extrapolate the movement of water vapor and reconstruct the gaps in multi-source microwave remote sensing water vapor observations. This method performs well in tracking large-scale weather systems, especially for tropical cyclones and atmospheric rivers. Advection methods also face

limitations due to simplified flow assumptions and source-sink interactions (Wimmers and Velden, 2011). Moreover, in complex terrain areas such as the TP, local circulations (such as valley winds and thermally-driven turbulence) have a significant impact on the water vapor transport path. Reanalysis wind fields often lack sufficient resolution to capture local circulations, leading to inaccurate extrapolation paths for water vapor transport over the TP. Overall, existing methods have made progress in improving temporal continuity. However, they remain limited by insufficient spatial resolution, which

hinders the detailed analysis of local water vapor transport processes. In addition, due to differences in observation times and the low temporal resolution of multi-source microwave sensors, systematic biases arise among datasets, often resulting in artifacts or inconsistencies in the fused TPW fields. In contrast, geostationary satellites (e.g., Himawari-8/9, H8/9) provide

stable, high-resolution TPW observations with spatial resolution around 2 km and temporal resolution of 10 minutes. Yet, current TPW fusion methods have not fully exploited the advantages offered by these observations.

To address the above limitations, this study proposes a novel TPW fusion framework that fully exploits the complementary strengths of microwave remote sensing's all-weather capability and the high spatiotemporal resolution of geostationary satellite observations, aiming to generate TPW data with both all-weather coverage and high resolution. Specifically, a virtual satellite constellation is constructed to jointly process data from eight polar- and inclined-orbit satellites and the Himawari-8/9 (H8/9) geostationary satellite. To address the systematic bias among multi-source microwave observations, a new correction method is developed that uses H8/9 TPW as the reference to calibrate the microwave retrievals from multiple satellites, enabling hourly fusion across data sources. In terms of spatial resolution, the effects of complex topography over the TP are considered by introducing surface elevation and related auxiliary information into the downscaling process, thereby enhancing the spatial resolution of the fused TPW data. In addition, an adaptive correction method is proposed to mitigate bias in high-resolution TPW data under cloudy conditions. Based on this framework, an hourly TPW dataset with a spatial resolution of 0.02° was generated over the TP for the period 2016–2022, and has been publicly released through the National Tibetan Plateau Data Center (https://doi.org/10.11888/Atmos.tpdc.301518, Ji et al., 2025b).

The remainder of this paper is organized as follows: Section 2 introduces the data sources and preprocessing steps; Section 3 describes the design and implementation of the fusion algorithm in detail; Section 4 validates the performance of the proposed algorithm and analyzes the characteristics of the generated dataset; Section 5 discusses the advantages of the algorithm and dataset and their application potential in meteorological and hydrological research; Section 6 summarizes the main achievements of this study and looks forward to future research directions.

## 2 Study area and data

### 2.1 Study area

The study area is the Tibetan Plateau, delineated by the vector boundary provided by Zhang. Y et al. (2021), with latitudinal and longitudinal extents ranging from 25° N to 40° N and from 67° E to 105° E, as shown in Fig. 1. This area is the highest and largest plateau in the world, with a relatively low atmospheric water vapor content. Moreover, influenced by the complex terrain and diurnal thermal changes, the water vapor in this area is easily blocked, accumulated and lifted, showing a strong local non-uniformity and vertical gradient (Xu et al., 2008). There are 44 GNSS stations distributed in this area, which can provide high-precision TPW data and offer reliable support for the accuracy verification of the multi-source remote sensing fusion TPW product in this study.

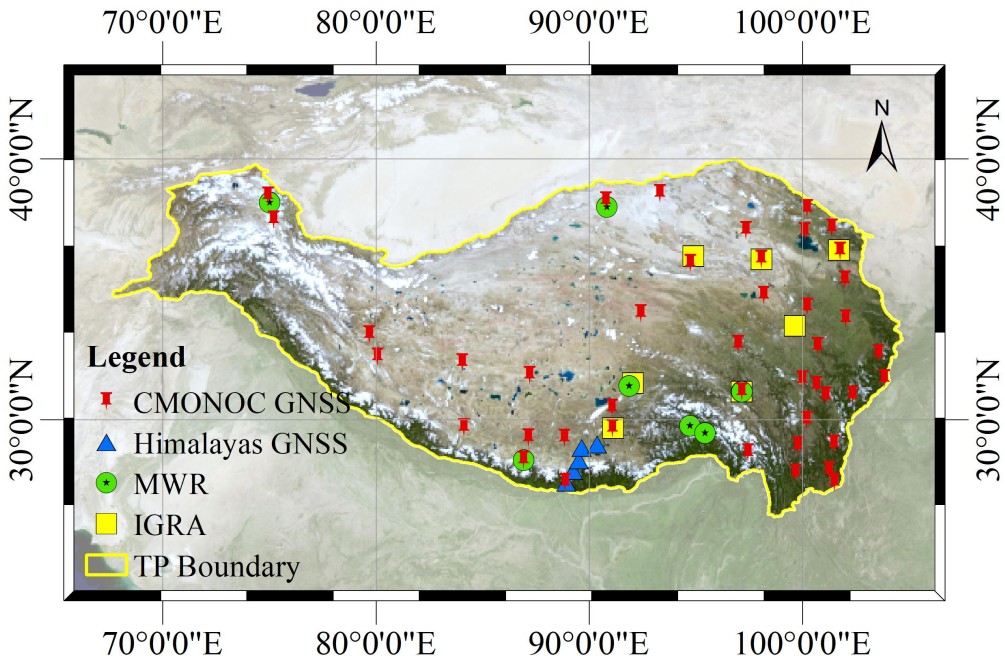

**Figure 1. Study area.**

## 2.2 Data

The datasets used for TPW fusion and validation in this study include TPW estimates from eight microwave remote sensing
satellites and from the Himawari-8/9 Advanced Himawari Imager (H8/9 AHI), GNSS TPW data, MIMIC-TPW2 data, ERA5 TPW data, and auxiliary datasets. Among them, remote sensing TPW data from the eight microwave satellites and H8/9 serve as key inputs for reconstructing all-weather TPW data with high spatial and temporal resolution. GNSS TPW data provide accurate ground-based water vapor observations and serve as reference data for validating the proposed fusion algorithm. MIMIC-TPW2 and ERA5 TPW are global water vapor products and are mainly used for comparison with the
TPW produced by our algorithm to evaluate its performance improvement. In addition, auxiliary data such as the vector boundary of the TP, elevation data, and spatiotemporal metadata are also employed. The details of each dataset are described below.

### 2.2.1 Microwave remote sensing TPW data from 8 satellites

The TPW data from eight microwave remote sensing satellites used in this study serve as key inputs for constructing all-
weather, high spatiotemporal resolution TPW data. These eight satellite datasets include TPW data from the Global Change Observation Mission – Water/Advanced Microwave Scanning Radiometer 2 (GCOM-W/AMSR2) and from the Microwave

Integrated Retrieval System (MIRS), both capable of providing coarse resolution total atmospheric water vapor data under all-weather conditions. The specific sources and characteristics of these two types of data are introduced below:

The AMSR2 TPW data, provided by our research team, is a global daily all-weather product with a 0.25° spatial resolution, and is freely available via the National Tibetan Plateau Data Center (https://cstr.cn/18406.11.Atmos.tpdc.272832, Ji et al., 2021, 2022). The atmospheric water vapor content over land is retrieved from AMSR2 brightness temperatures at 18.7 and 23.8 GHz using the improved TPW retrieval algorithm developed by Ji et al. (2017). Verified by Suomi GPS TPW, the root mean square error (RMSE) of AMSR2 TPW data over global land is in the range of 3.5－5.2 mm. In this study, AMSR2 TPW data over the TP were extracted using the vector boundary defined by Zhang. Y et al. (2021), and primarily serve as all-weather inputs for the fusion algorithm.

In addition, the microwave remote sensing TPW data from MIRS includes data from seven satellites, namely National Oceanic and Atmospheric Administration-18/19 (NOAA-18/19), Meteorological Operational Satellite-A/B (MetOp-A/B), Defense Meteorological Satellite Program-F17/18 (DMSP-F17/18), and Global Precipitation Measurement Core Observatory (GPM core). Among them, NOAA-18/19 and MetOp-A/B are equipped with Advanced Microwave Sounding Unit (AMSU) and Microwave Humidity Sounder (MHS) payloads, DMSP-F17/18 is equipped with Special Sensor Microwave Imager/Sounder (SSMIS) payload, and GPM core is equipped with GPM Microwave Imager (GMI) payload. MIRS utilizes all available channels from these instruments, including water vapor–sensitive frequencies near 22.235 GHz and 183 GHz. These channels respond to water vapor at different altitudes, enabling MIRS to retrieve atmospheric humidity profiles effectively—even under cloudy conditions. The MIRS TPW is calculated by vertically integrating the retrieved humidity profiles. Validation against IGRA radiosonde data shows that MIRS TPW data retrieved from different sensors exhibit RMSEs of 5.03–5.94 mm over land areas (Boukabara et al., 2010). Table 1 summarizes the microwave sensors used in MIRS TPW retrieval and their local overpass times across the TP, which typically range from 1 to 3 hours apart. Users can access and download MIRS TPW data from NOAA's Comprehensive Large Array-data Stewardship System (https://www.aev.class.noaa.gov/saa/products/search?sub_id=0&datatype_family=MIRS_ORB). As with the AMSR2 TPW data, the MIRS TPW data were preprocessed and used as key inputs to provide coarse-resolution, hourly, all-weather water vapor information in the fusion algorithm.

**Table 1. Microwave Remote Sensing TPW Data for the Fusion Algorithm (March 1, 2017)**

| Serial Number | Sensor Name | Sensor Type | Satellite Platform | Observation Time (local time) | Spatial Resolution |
|---|---|---|---|---|---|
| 1 | AMSR-2 | Microwave Radiometer | GCOM-W | 01:30p.m. (descending node, polar orbit) | 0.25˚ |
| 2 | AMSU+MHS | Microwave Sounder | NOAA-18 | 04:50p.m. (descending node, polar orbit) | 17km |
| 3 | AMSU+MHS | Microwave Sounder | NOAA-19 | 06:50p.m. (descending node, | 17km |

| | | | | polar orbit) | |
|---|---|---|---|---|---|
| 4 | AMSU+MHS | Microwave Sounder | MetOp-A | 06:46p.m. (descending node, polar orbit) | 17km |
| 5 | AMSU+MHS | Microwave Sounder | MetOp-B | 07:46p.m. (descending node, polar orbit) | 17km |
| 6 | SSMIS | Microwave Sounder | DMSP-F17 | 08:46p.m. (descending node, polar orbit) | 17km |
| 7 | SSMIS | Microwave Sounder | DMSP-F18 | 09:30p.m. (descending node, polar orbit) | 17km |
| 8 | GMI | Microwave Imager | GPM core | --(Inclined orbit) | 9km |

**2.2.2 Himawari-8/9 geostationary satellite infrared remote sensing TPW data**

Currently, the H8/9 geostationary satellite does not provide official TPW product. This study adopted a neural network-based rapid retrieval algorithm for TPW over the TP proposed by Jiang. J et al. (2022). Based on this algorithm, a high accuracy TPW dataset was produced under clear-sky conditions over the region spanning 80°E–105°E and 25°N–40°N from 2016 to 2022, with hourly temporal and 0.02° spatial resolution (https://cstr.cn/18406.11.Atmos.tpdc.301522, Wang and Liu, 2024). The retrieval algorithm uses not only traditional split-window channels but also optimizes the channel combination by selecting water vapor sensitive bands from the H8/9 AHI sensor (specifically 11.2 μm, 12.3 μm, 7.0 μm, and 7.3 μm), thereby improving retrieval accuracy in low-moisture conditions (TPW < 2 cm) and over complex surfaces. During data generation, a 24-hour cloud detection algorithm developed by Shang et al. (2024) based on the characteristics of H8/9 channels was integrated to provide precise cloud masks, thereby minimizing the impact of inaccurate cloud identification on the accuracy of TPW inversion under clear-sky conditions. To improve computational efficiency, a neural network model was introduced to accelerate radiative transfer calculations. Verified against GNSS TPW data, the RMSE of this dataset under clear-sky conditions over the TP is approximately 2 mm, with a correlation coefficient of 0.95, demonstrating its high accuracy. In this study, the H8/9 TPW product was fused with TPW data retrieved from eight microwave remote sensing satellites, serving as a key source of high-resolution, clear-sky water vapor information for all-weather TPW reconstruction over the TP.

**2.2.3 GNSS TPW data**

The GNSS TPW data used in this study were generated based on the retrieval method proposed by Zhang. H et al. (2019). This method relies on continuous GNSS observations from the CMONOC. By utilizing GNSS signals and surface pressure, Zenith Total Delay (ZTD) is derived through real-time Precise Point Positioning (PPP), while Zenith Hydrostatic Delay (ZHD) is calculated using the Saastamoinen model. The zenith wet delay (ZWD), which is related to TPW, is then derived as the difference between ZTD and ZHD (ZWD = ZTD−ZHD). To convert ZWD into TPW, Zhang. H et al. (2019) introduced

the Gridded-Mixed Tm (GM-Tm) model, which estimates the weighted mean temperature (Tm) based on surface temperature, station coordinates, and day of year. The conversion factor $\Pi$ is computed as a function of Tm, and TPW is then calculated using the relationship TPW = $\Pi$ × ZWD. This method avoids the need for vertical temperature profiles and enables accurate and real-time TPW retrieval from GNSS observations. Moreover, the method also incorporates a parallel computing framework to support real-time TPW production at 215 GNSS stations across China, with a temporal resolution of 5 minutes. Validation results show that the retrieved TPW has root mean square errors (RMSEs) of 1.7 mm and 2.0 mm when compared with radiosonde-derived TPW (RS-TPW) and NCEP-II reanalysis TPW (NCEP-II-TPW), respectively, indicating good accuracy and stability. In this study, TPW data for 44 GNSS stations over the TP in 2017 were generated using this method and used to validate the results from the multi-source fusion algorithm.

### 2.2.4 IGRA radiosonde TPW data

The Integrated Global Radiosonde Archive (IGRA) provides vertical profiles of temperature, humidity, and pressure at multiple levels. In this study, the TPW derived from IGRA observations in 2022 was used to independently validate the fused TPW dataset. The IGRA data were obtained from the National Climatic Data Center (Durre et al., 2006).

### 2.2.5 Scientific Expedition ground-based TPW observations

Two types of ground-based observations from the scientific expedition were employed to further evaluate the fused TPW data:

(1) Microwave radiometer (MWR) observations: Continuous tropospheric observations (0 – 10 km) were obtained from the MWR network deployed across the Plateau during 2021 – 2022 (Chen & Ma, 2022; Chen et al., 2024). The instruments provide profiles of temperature and humidity at 58- or 83-layer vertical resolutions, from which total precipitable water (TPW) is derived. Observation sites include MAWORS, NADOR, Mangai, Naqu, Changdu, Leshan, QOMS, SETS, and Kabu.

(2) GNSS TPW observations in the central Himalayas: A five-station north – south GNSS chain was established along the Yadong – Lhasa fault zone (XYDX, DNLG, SMDX, JZLZ, and LKZZ), providing high-accuracy ( ± 0.1 mm) TPW retrievals from 2015 to 2019 (Wang et al., 2019; Yang, 2023). These data were used to validate the fused TPW performance in steep terrain regions of the central Himalayas.

### 2.2.4 MIMIC-TPW2 data

The MIMIC-TPW2 data, released by the Cooperative Institute for Meteorological Satellite Studies (CIMSS) at the University of Wisconsin-Madison, is a global, hourly, and 0.25° TPW product generated by fusing MIRS TPW data from multiple microwave sensors. The core algorithm of this data is the advection fusion method proposed by Wimmers and Velden (2011). This method simulates water vapor advection trajectories using the NCEP GFS wind field and fuses MIRS

TPW data via extrapolation. In this study, this data is used for lateral comparison with the TPW fused using the algorithm in this study. The MIMIC-TPW2 data can be obtained from https://bin.ssec.wisc.edu/pub/mtpw2.

### 2.2.5 ERA5 TPW data

ERA5 TPW data, released by the European Centre for Medium-Range Weather Forecasts (ECMWF), is a widely used global reanalysis product. It is based on the Integrated Forecast System (IFS) Cycle 41r2 and the four-dimensional variational (4D-Var) assimilation framework, integrating multi-source observational data to generate hourly, 0.25° resolution global water vapor fields. As an important benchmark for climate research, ERA5 TPW has been applied in studies of water vapor transport (Sun et al., 2022), the relationship between precipitation and water vapor budget (Wu et al., 2023), and the diagnosis of extreme precipitation events over the TP (Chen et al., 2020). This study compares ERA5 TPW with new algorithm products to assess the improvement potential of the latter in the highly heterogeneous region of the TP. The data can be accessed through the ECMWF Climate Data Store (https://cds.climate.copernicus.eu/datasets/reanalysis-era5-single-levels?tab=overview).

### 2.2.6 Auxiliary data

The auxiliary data used in this study mainly include the vector boundary of the TP, elevation data, and geographic coordinates and time. The vector boundary was derived by Zhang. Y et al. (2021) through analysis of the Advanced Spaceborne Thermal Emission and Reflection Radiometer Global Digital Elevation Model (ASTER GDEM) and Google Earth imagery, and was used to extract multi-source remote sensing TPW data within the study area. Elevation data were obtained from the Earth Topography 2022 (ETOPO2022) surface elevation dataset with a spatial resolution of 15 arc-seconds, provided by NOAA (NOAA National Centers for Environmental Information, 2022). In addition, geographic coordinates, time, and elevation were used as auxiliary variables to represent longitudinal and latitudinal variation, diurnal cycles, and vertical terrain effects of water vapor, and were further employed for error correction and downscaling modeling of the fused TPW data.

### 3 Methodology

To achieve all-weather, high-resolution atmospheric water vapor fusion, this study constructed a virtual satellite constellation composed of eight microwave remote sensing satellites (including GCOM-W, NOAA-18/19, MetOp-A/B, DMSP-F17/18, and GPM Core) and H8/9 geostationary meteorological satellite. This virtual constellation integrates the observational advantages of different types of remote sensing data: the microwave remote sensing data from multiple satellites can provide global coverage and all-weather coarse-resolution TPW data with a time interval of approximately 3 hours; while H8/9 provide stable and continuous TPW data with a 2-km spatial resolution every hour under clear-sky conditions. By fully integrating the advantages of the two types of remote sensing observations, a set of high-precision TPW fusion algorithms

suitable for complex terrain and cloudy conditions was developed, including key technologies such as systematic bias calibration among multi-source microwave remote sensing TPW, spatial downscaling of coarse-resolution TPW, and adaptive correction of high-resolution TPW under cloudy conditions. Based on these techniques, the all-weather TPW data over the TP can be generated, featuring 0.02° spatial and hourly temporal resolution. The detailed algorithmic framework and implementation steps are described below.

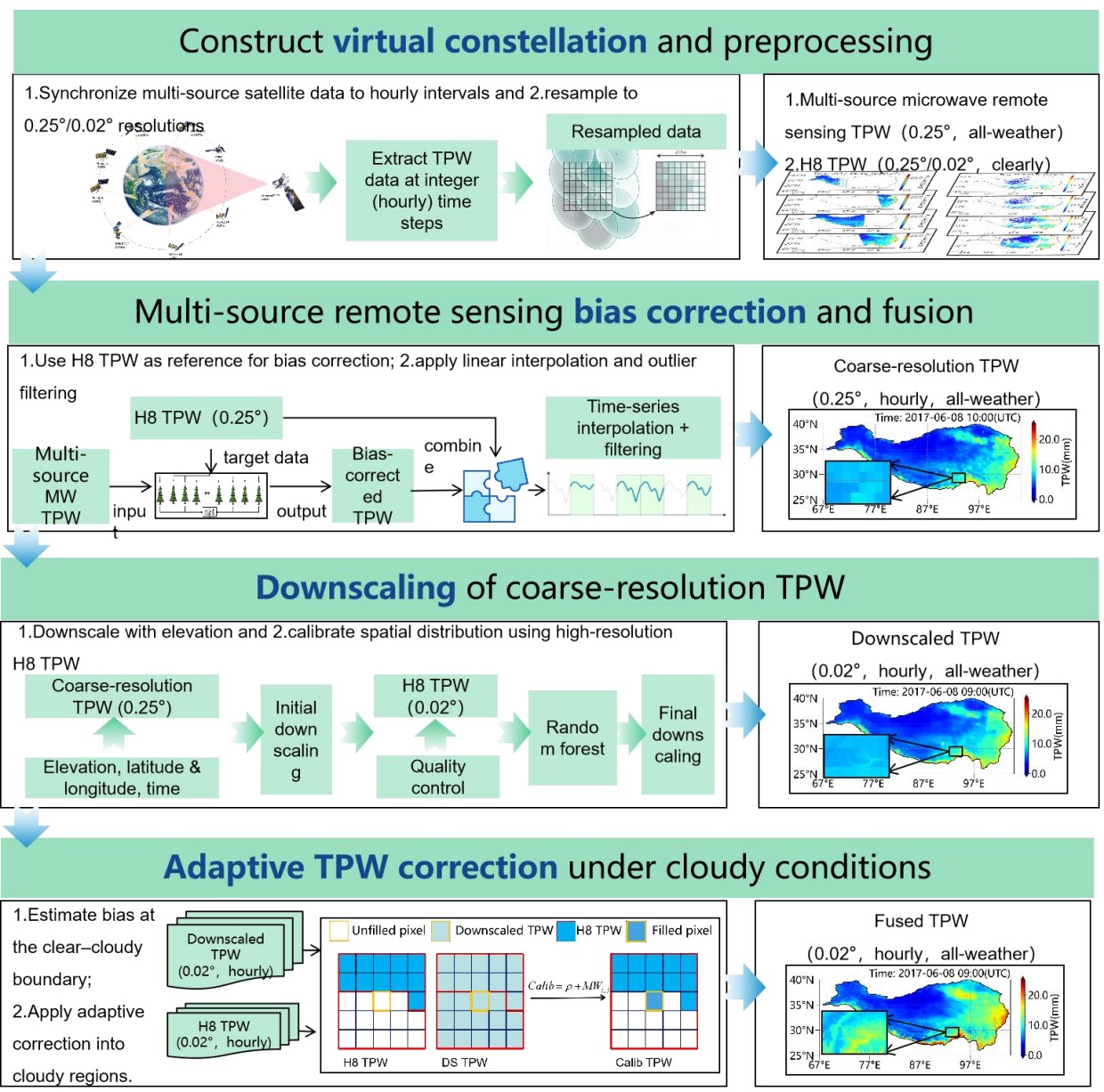

**Figure 2. Flowchart of the multi-source remote sensing TPW fusion algorithm**

### 3.1 Construction of the virtual satellite constellation

The concept of a virtual satellite constellation was proposed by the Committee on Earth Observation Satellites (CEOS) in 2005 as a technical framework for enhancing Earth system monitoring by integrating observations from multiple satellite platforms. Its core idea lies in combining the strengths of different orbits and sensor types to form a synergistic observation network with broad coverage and high spatiotemporal resolution. This study selected microwave remote sensing TPW data from eight polar and inclined orbit satellites, including DMSP-F17/F18, NOAA-18/19, Metop-A/B, GCOM-W1, and GPM. The microwave payloads of these satellites have global coverage and all-weather observation capabilities. Meanwhile, high-resolution clear-sky observation data from H8/9 AHI was utilized to compensate for the deficiencies of microwave observations in regional details. By constructing a virtual constellation and processing multi-source satellite data collaboratively, input data with uniform spatiotemporal resolution were provided for subsequent fusion algorithms.

In this study, the construction of the virtual constellation aimed to address the consistency issues of multi-source satellite data in terms of observation time and spatial resolution. In terms of spatial uniformity, based on the latitude and longitude information of the observation pixels of each satellite, bilinear interpolation was used to reproject the multi-source data onto 0.25° and 0.02° equal latitude and longitude grids. Among them, the multi-source microwave remote sensing TPW data were uniformly reprojected onto a 0.25° grid to ensure consistent spatial resolution among different data sources. H8/9 TPW data, which provide stable high-resolution observations, were first reprojected onto a 0.02° grid to provide high-resolution water vapor information and then resampled to a 0.25° grid to serve as a reference for calibrating algorithm errors and time deviations among multi-source microwave remote sensing TPW data. In terms of temporal uniformity, the data were divided into hourly intervals based on the actual UTC overpass times of each satellite. Data within 30 minutes before and after each hour were assigned to that hour, resulting in the construction of hourly microwave remote sensing TPW data. Taking the hourly matching results of different satellites over the TP on March 1, 2017, as an example (Fig. 3), the maximum difference in scanning times among the satellites was approximately 3 hours. The spatiotemporal distribution of the multi-source microwave remote sensing TPW data after hourly mosaicking is shown in Fig. 4(a). In contrast, the H8/9 satellite continuously observes the entire disk region at 10-minute intervals. By directly extracting the observation data at the integer hour, hourly H8/9 TPW data can be generated. The spatiotemporal distribution of the processed H8/9 TPW data is shown in Fig. 4(b). In the fusion framework, the hourly H8/9 TPW data provide the baseline observations. At time steps without microwave data, the hourly TPW fields are derived from H8/9 and subsequently reconstructed through spatiotemporal interpolation under cloudy conditions. When multi-source microwave data are available, they are bias-corrected and used to fill the cloudy gaps in the H8/9 retrievals. If missing values remain, the reconstruction step was repeated until full hourly coverage was achieved.

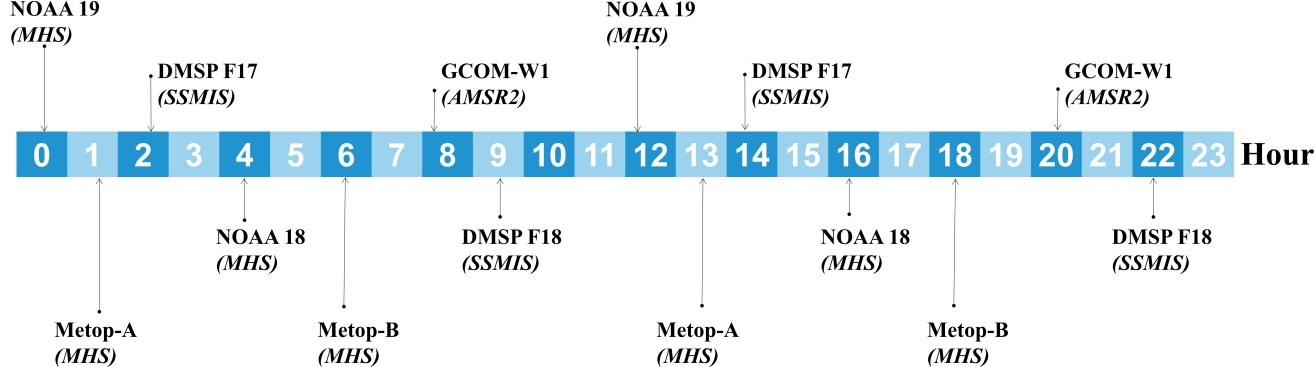

**Figure 3. Hourly matching results of multi-source microwave remote sensing satellites over the Tibetan Plateau on March 1, 2017 (UTC time)**

## 3.2 Bias correction and fusion of multi-source TPW at coarse resolution

Due to differences in sensor types, orbital configurations, and retrieval algorithms, TPW data from multi-source microwave remote sensing satellites exhibit significant systematic biases, which affect the consistency and accuracy of the fused product. To effectively integrate TPW observations from various satellites, it is necessary to perform bias correction and data reconstruction to eliminate inter-sensor biases and temporal inconsistencies and to ensure the generation of spatiotemporally continuous TPW fields. Among the satellite data used, H8/9 TPW data can provide stable hourly observations and have advantages in temporal resolution and observation consistency. Therefore, this study selects it as the reference and performs bias correction on the reprojected multi-source microwave remote sensing TPW data at a resolution of 0.25°. After bias correction, multi-source microwave remote sensing data were used to fill the observation gaps of H8/9 under cloudy conditions. However, the hourly TPW data after filling still have some spatial missing data, especially in areas with large-scale missing data of H8/9 due to severe cloud cover. Therefore, interpolation and filtering methods were further introduced to complete the remaining missing data areas, resulting in the spatiotemporally continuous, hourly, 0.25° resolution coarse TPW data, which serves as the input for subsequent high-resolution TPW reconstruction. The specific operations are as follows:

Bias calibration of multi-source microwave remote sensing TPW data. Considering the advantages of H8/9 data, such as high-quality observation data, high spatiotemporal resolution, and stable observation accuracy, it is selected as the reference for correcting microwave remote sensing TPW data. To address the temporal and algorithmic discrepancies among multi-source microwave data, this study employs the random forest algorithm (Breiman, 2001) to develop a correction model. The target data is H8/9 TPW data under clear-sky conditions, and the model training input data include microwave remote sensing TPW data, longitude, latitude, elevation, and time, etc. The model was trained and updated separately for each satellite and for each month of every year to account for seasonal variations. For example, the correction model for NOAA-19 in January 2017 was trained using 11,565 valid collocated clear-sky samples with H8 TPW. The correction model was

constructed under clear-sky conditions and applied to all-weather conditions. This month-by-month, satellite-specific training ensures that systematic bias correction was dynamically adapted to both temporal and sensor-related variations.

Reconstruction of coarse-resolution hourly TPW data in missing areas. Due to the spatial coverage gaps between microwave
remote sensing satellite orbits, multi-source microwave remote sensing observations cannot provide effective coverage in every hour in the same area. Meanwhile, H8/9 infrared remote sensing observations from geostationary satellites have a large number of missing data under cloud cover. To address these missing data, the bias-corrected multi-source microwave remote sensing TPW data were used to fill the missing data of H8/9 under cloud cover. The spatiotemporal distribution of the filled TPW data is shown in Fig. 4(c). The spatiotemporal coverage of the filled data has been greatly improved.
Although there are still some areas with missing data, the situation of long-term consecutive missing data in the same area has been effectively avoided. For the remaining missing regions after mosaicking, a combined temporal–spatial interpolation strategy was adopted to achieve continuous hourly coverage. Specifically, linear interpolation was applied along the time dimension between two adjacent valid observations when the temporal gap was less than or equal to 24 hours, while longer gaps or edge segments remained missing. Missing pixels in each hourly 2D TPW field were then interpolated
spatially using a bilinear method, but only when more than 50% of the pixels within the surrounding $1°×1°$ neighborhood were valid. Pixels lacking sufficient valid neighbors were left unfilled. When residual missing values remained after spatial interpolation, the temporal and spatial interpolation procedures were iteratively repeated until no further gaps were present. Finally, a Savitzky–Golay filter was applied to smooth local variations in the reconstructed field, resulting in hourly, 0.25° resolution fused TPW data. The reconstruction result is shown in Fig. 4(d). The percentage of multi-source remote sensing
observations covering the TP at the hourly scale in 2017 was statistically analyzed (Fig. 4(e)). The average coverage rate of multi-source microwave remote sensing TPW data were 27.2%, and that of H8/9 TPW data were 20.1%. After mosaicking, the overall coverage rate increased to 41.7%, and after interpolation and reconstruction, the total coverage rate reached 100%. Compared to single-satellite observations, this study successfully overcame the spatiotemporal limitations inherent in single-satellite observations by integrating data from geostationary satellites and multi-source microwave remote sensing satellites,
thereby significantly enhancing the spatial and temporal coverage.

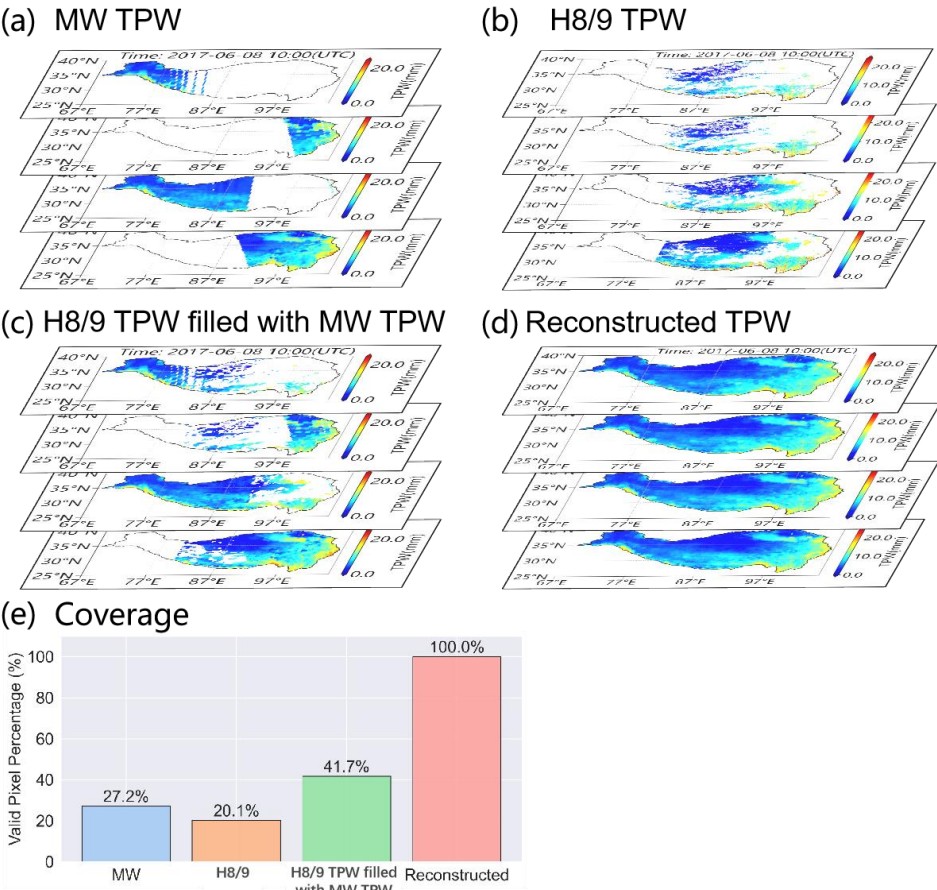

**Figure 4. Spatial and temporal distribution of multi-source remote sensing TPW data over the TP (a. Multi-source microwave remote sensing mosaicked TPW data; b. H8/9 TPW data; c. Mosaicked TPW data from multi-source microwave remote sensing and H8/9; d. Coarse-resolution reconstructed TPW data; e. Hourly multi-source remote sensing observation coverage percentage over the TP in 2017)**

### 3.3 Downscaling of coarse-resolution TPW

Although the coarse-resolution hourly TPW data derived from the previous multi-source fusion can achieve all-weather coverage, its relatively low spatial resolution limits its ability to capture the local variation characteristics of water vapor over the complex terrain of the TP. Especially in valleys and slopes, the coarse-resolution data can easily mask the influence of terrain on water vapor distribution. To enhance the spatial resolution of the reconstructed TPW data in the topographically complex regions, this study introduced auxiliary information such as elevation, longitude, latitude, and time to conduct a preliminary downscaling of the coarse-resolution reconstructed TPW data. On this basis, the preliminary results were further corrected by combining the high-resolution observations of H8/9 under clear-sky conditions.

Preliminary spatial downscaling of coarse-resolution reconstructed TPW data. To enhance the spatial resolution of coarse-resolution reconstructed TPW data in regions with significant topographic variation, a spatial downscaling model was

developed using the random forest algorithm. The 0.25° coarse-resolution reconstructed TPW data served as the target variable, while elevation, latitude, longitude, and time were used as input features to account for the effects of spatial location, topographic characteristics, and temporal variation on TPW distribution. The model was trained using input and output data at the 0.25° resolution, and subsequently applied to input variables at 0.02° resolution to produce preliminary hourly high-resolution TPW data on all-weather conditions. Following the temporal sampling strategy proposed by Sun et al. (2024), training samples were selected within a ±3-hour window centered on each target hour. For instance, on 1 January 2017, the DEM-based downscaling model was trained using 27,360 valid samples.

Refining the preliminary downscaling with H8/9 high-resolution TPW data. Although the spatial resolution of the preliminary downscaling results was improved, there might still be spatial detail deviations due to the lack of actual high-resolution water vapor observation information. Therefore, the H8/9 TPW data under clear-sky conditions was further introduced as a reference to correct the preliminary downscaling results. Before constructing the correction model, quality control was conducted on the training data samples. Specifically, the differences between the preliminarily downscaled TPW and H8/9 TPW were calculated, and only the samples with differences within ±3 standard deviations from the mean were retained. Subsequently, with H8/9 TPW as the target variable and the preliminary downscaling TPW as the input variable, combined with auxiliary information such as longitude, latitude, elevation, and time, a random forest algorithm was used to construct the correction model. This model was trained under clear-sky conditions and applied to all-weather conditions to correct the spatial distribution of the preliminary downscaled TPW data, obtaining hourly, 0.02° resolution downscaling TPW data (DS TPW). For instance, on 1 January 2017, the H8-based correction model was trained using 487,986 valid samples.

## 3.4 Adaptive correction of high-resolution TPW under cloudy conditions

The previous step has reconstructed the all-weather hourly, 0.02° resolution DS TPW data. Since the existing correction model was developed under clear-sky conditions, and water vapor distributions differ between clear and cloudy skies, the DS TPW values may exhibit biases when applied under cloudy conditions. This deviation mainly stems from the limited applicability of the model relationship established under clear-sky conditions to cloudy conditions. Consequently, it is necessary to introduce a further correction mechanism to enhance the accuracy of TPW data under cloudy conditions. Based on DS TPW and H8/9 TPW, this study developed an adaptive correction method for high-resolution TPW by exploiting boundary differences between clear and cloudy sky, thereby improving accuracy under cloudy conditions. This method assumes spatiotemporal continuity in the water vapor field and uses H8/9 TPW observations from clear-sky boundary areas to dynamically estimate regional bias, which is then applied for pixel-wise adaptive correction of DS TPW under cloudy conditions.

This adaptive correction begins at the boundary between clear-sky and cloudy regions. A 5×5 window is used to select H8/9 and DS TPW data, and the mean values of valid pixels (values greater than 0) are calculated respectively to construct the regional deviation termed as follows:

$$\delta_{i,j} = \mu_{i,j}^{H8/9} - \mu_{i,j}^{DS} \tag{1}$$

where (i,j) is the pixel position to be corrected under the cloudy conditions, $\mu_{i,j}^{H8/9}$ and $\mu_{i,j}^{DS}$ represent the mean values of the valid pixels within the 5×5 window centered at pixel (i,j) in H8/9 TPW and DS TPW, respectively. The calculated δ reflects the systematic deviation of H8/9 TPW data from DS TPW data at the clear-sky boundary. By applying the calculated deviation term δ to correct the DS TPW at the pixel position under the cloudy conditions, the correction formula is as follows:

$$TPW_{i,j}^{calib} = \delta_{i,j} + TPW_{i,j}^{DS} \tag{2}$$

In this equation, $TPW_{DS}$ refers to the original downscaled TPW value at the cloudy pixel, and $TPW_{Calib}$ is the corrected TPW. The correction process starts from the clear-sky boundary and gradually extends into the cloud region, dynamically computing local bias and applying correction at each pixel. This method has adaptive capabilities, allowing for flexible determination of the correction magnitude based on the observation conditions of different regions, thereby improving the spatial consistency and accuracy of TPW data under cloudy conditions. Finally, the corrected high-resolution TPW data under cloudy conditions were used to fill missing areas in the H8/9 TPW dataset, resulting in hourly fused TPW data with a spatial resolution of 0.02°.

### 3.5 Dataset

Based on the aforementioned multi-source data fusion method, this study produced a TPW dataset over the TP using TPW retrievals from eight polar and inclined orbit satellites and the H8/9 geostationary satellite. The dataset covers the period from 2016 to 2022 and features hourly temporal resolution and 0.02° spatial resolution with all-weather coverage. The dataset has been published by the National Tibetan Plateau Data Center and is available at: https://doi.org/10.11888/Atmos.tpdc.301518 (Ji et al., 2025b). This product can be applied to improve studies on water vapor transport, precipitation forecasting, and data assimilation in numerical models over the TP region.

### 3.6 Evaluation metrics

To evaluate the accuracy of the fused TPW dataset, ground-based GNSS TPW observations from the CMONOC stations across TP were used as reference data. The validation focuses on four statistical indicators: the correlation coefficient (R), bias, root mean square error (RMSE), and relative root mean square error (RRMSE). These metrics are calculated using the following equations:

$$R = \frac{\sum_{i=1}^{N}(x_i - \mu_x)(y_i - \mu_y)}{\sqrt{\sum_{i=1}^{N}(x_i - \mu_x)^2} \cdot \sqrt{\sum_{i=1}^{N}(y_i - \mu_y)^2}} \tag{3}$$

$$Bias = \frac{1}{N}\sum_{i=1}^{N}(x_i - y_i) \tag{4}$$

$$RMSE = \sqrt{\frac{1}{N}\sum_{i=1}^{N}(x_i - y_i)^2} \tag{5}$$

$$RRMSE = \frac{RMSE}{\mu_y} \times 100\% \tag{6}$$

In Eqs. (3)-(6), $x_i$ and $y_i$ denote the $i$-th values of the fused TPW and the GNSS TPW, respectively; $\mu_x$ and $\mu_y$ represent their
corresponding mean values; and $N$ is the total number of matched sample pairs used in the evaluation.

## 4 Results

### 4.1 Accuracy verification of fusion results

#### 4.1.1 Validation of TPW under clear-sky and cloudy conditions

To evaluate the performance of the fusion algorithm under different sky conditions, the 2017 GNSS TPW observations were
used to validate the H8 TPW, the multi-source MW TPW, and the fused TPW data. During validation, each GNSS station
was matched with the nearest remote sensing grid cell, and GNSS measurements at hourly timestamps were used to validate
the corresponding fused TPW. The corresponding results are summarized in Tab. 2. Under clear-sky conditions, the H8
TPW shows the highest correlation coefficient of 0.95, followed by 0.89 for the MW TPW. The biases are 0.38 mm for H8
and −1.82 mm for MW. The RMSE values are 1.94 mm for H8 and 3.79 mm for MW, confirming that the H8 TPW product
provides the most reliable estimates under clear-sky conditions and justifying its use as the reference for bias correction.
Under cloudy conditions, the fused TPW maintains a correlation coefficient of 0.95, higher than 0.88 for the MW TPW. The
biases are −2.78 mm for the fused TPW and −4.31 mm for MW, and the RMSE values are 4.92 mm and 6.44 mm,
respectively. These results demonstrate that the fusion algorithm effectively reduces bias and error under cloudy conditions,
leading to a notable improvement in accuracy compared with the original MW observations.
**Table 2. Comparison of hourly TPW from H8 TPW, MW TPW, and fused TPW against GNSS observations in 2017.**

| Weather conditions | Data type | R | Bias (mm) | RMSE (mm) | RRMSE (%) | N |
|---|---|---|---|---|---|---|
| Clear sky | H8 TPW | 0.95 | 0.38 | 1.94 | 27.24 | 143670 |
| | MW TPW | 0.89 | -1.82 | 3.79 | 54.37 | 42367 |
| Cloudy | MW TPW | 0.88 | -4.31 | 6.44 | 55.44 | 46545 |
| | Fused TPW | 0.95 | -2.78 | 4.92 | 42.32 | 142063 |

**4.2.2 Comparison with other TPW products**

To evaluate the fusion algorithm, GNSS TPW data from 2017 were selected as reference, and the fused TPW product generated for the same year was validated under all-weather conditions at three temporal scales: hourly, daily, and monthly. The "hourly" scale refers to direct validation of the hourly fused TPW data, while the "daily" and "monthly" scales represent validations based on daily- and monthly-averaged TPW values derived from the hourly product. In addition, ERA5 reanalysis data and the multi-source remote sensing fusion product MIMIC-TPW2, both of which provide hourly TPW estimates, were included for comparative validation. Figure. 5(a)-(c) show the hourly validation results. The RMSE of the fused TPW is 3.79 mm, which is 10.82% and 6.19% lower than that of MIMIC-TPW2 (4.25 mm) and ERA5 (4.04 mm), respectively. The Bias of the fused TPW is −1.15 mm, smaller than that of MIMIC-TPW2 (−2.02 mm) and ERA5 (−2.22 mm). Figure. 5(d)-(f) show the verification results at the daily scale. The RMSE of the fused TPW drops to 3.50 mm, which is 10.26% and 9.60% lower than that of MIMIC-TPW2 (3.90 mm) and ERA5 (3.87 mm), respectively. Figure. 5(g)-(i) show the verification results at the monthly scale. The RMSE of the fused TPW reduces to 3.19 mm, which is 12.84% and 15.16% lower than that of MIMIC-TPW2 (3.66 mm) and ERA5 (3.76 mm), respectively.

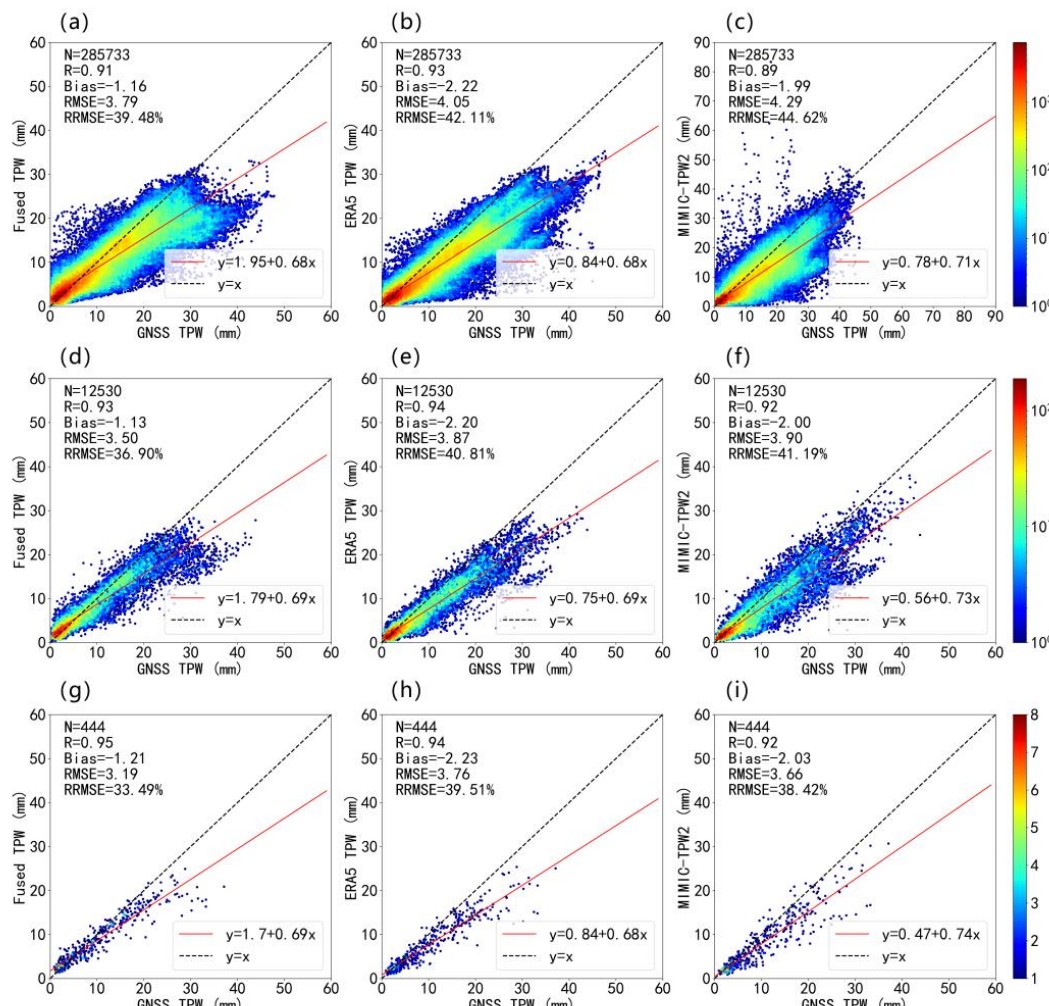

**Figure 5. Scatterplot comparisons of GNSS TPW with fused TPW, ERA5 TPW, and MIMIC-TPW2 under all-sky conditions at hourly (first row), daily (second row), and monthly (third row) scales. ( a, d, g: fused TPW;b, e, h: ERA5 TPW;c, f, i: MIMIC-TPW2 TPW)**

The previous text indicates that the overall accuracy of the fused TPW is superior to that of ERA5 TPW and MIMIC-TPW2 at different temporal scales. However, due to the complex terrain of the TP, where water vapor exhibits substantial regional heterogeneity, it is essential to evaluate whether the fused data maintains stable and reliable performance across different subregions. Therefore, station-scale validation is conducted by comparing the fused TPW with ERA5 TPW and MIMIC-TPW2.

Figure 6 presents the distributions of RMSE and RRMSE for the fused TPW, ERA5 TPW, and MIMIC-TPW2 across GNSS stations under all-weather conditions. The left column (Fig. 6(a), (c), (e)) shows the RMSE, while the right column (Fig. 6(b),

(d), (f) shows the RRMSE, with the three rows corresponding to the fused TPW, ERA5 TPW, and MIMIC-TPW2, respectively. As shown in the Fig. 6, the fused TPW exhibits relatively lower RMSE at most stations, particularly in the southeastern region. In contrast, ERA5 TPW and MIMIC-TPW2 display larger errors in areas with sparse stations and at high-elevation sites in the western Plateau. In terms of RRMSE, the fused TPW shows lower errors in southern and northeastern parts of the TP. Particularly in station-sparse areas such as the Himalayas, ERA5 TPW and MIMIC-TPW2 exhibit larger RRMSE. By comparison, the fused TPW, which combines geostationary and multi-source microwave remote sensing observations, consistently delivers low-error and stable water vapor estimates even in regions with sparse stations and complex terrain.

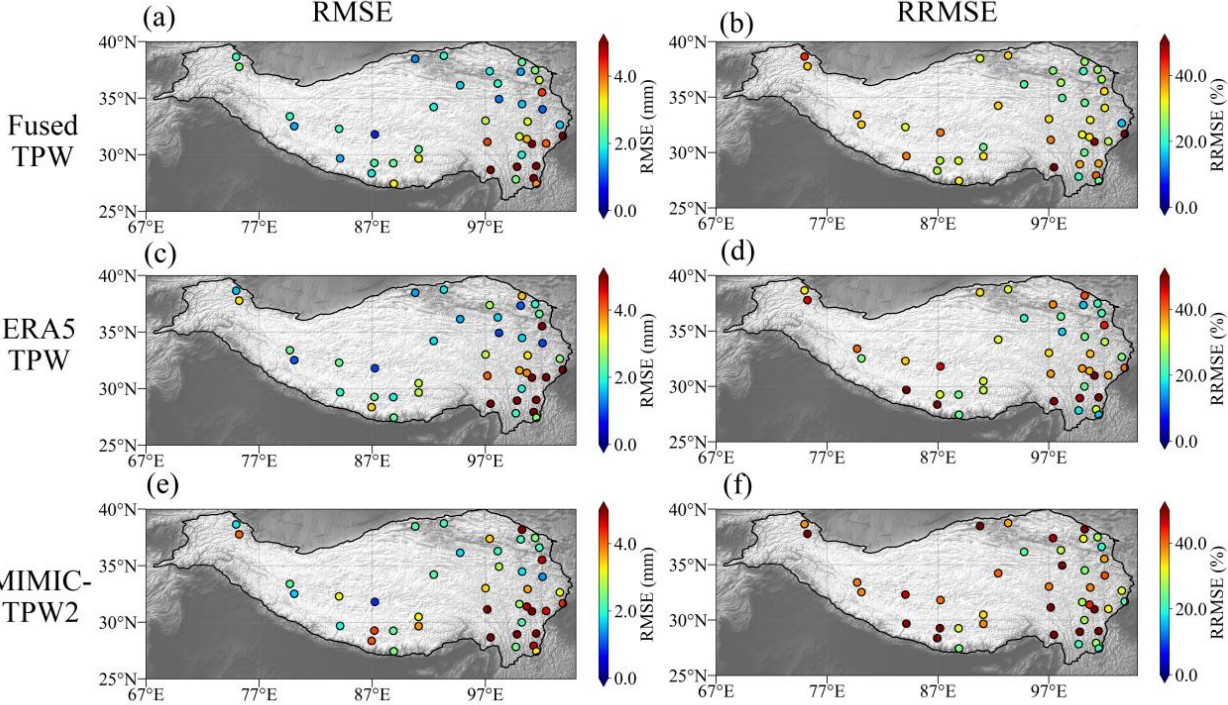

**Figure 6. Comparison of RMSE (first column) and RRMSE (second column) at the station scale for the fused TPW, ERA5 TPW, and MIMIC-TPW2 under all-weather conditions. (a,b: fused TPW; c, d: ERA5 TPW; e, f: MIMIC-TPW2)**

### 4.1.3 Additional validation using radiosonde and scientific expedition data

To further evaluate the performance of the fused TPW dataset, additional validations were conducted using (1) IGRA radiosonde TPW data from 2022, (2) microwave radiometer (MWR) observations obtained from the scientific expedition over the Tibetan Plateau in 2022, and (3) GNSS TPW observations from the central Himalayas in 2017. Table.3 shows the metrics comparisons between the fused TPW, ERA5, and MIMIC-TPW2.

**Table 3. Validation of the fused TPW, ERA5 TPW, and MIMIC-TPW2 products using independent observations from IGRA, ground-based MWR, and GNSS over the central Himalayas.**

| Observation Type | Data Type | R | Bias (mm) | RMSE (mm) | RRMSE (%) | N |
|---|---|---|---|---|---|---|
| IGRA (2022) | Fused TPW | 0.96 | 0.08 | 2.15 | 27.66 | 5059 |
| | ERA5 TPW | 0.98 | −0.34 | 1.55 | 19.93 | 5059 |
| | MIMIC-TPW2 | 0.89 | −0.38 | 3.26 | 41.93 | 5059 |
| MWR (2022) | Fused TPW | 0.86 | −3.63 | 6.34 | 68.06 | 19280 |
| | ERA5 TPW | 0.92 | −4.35 | 6.21 | 66.63 | 19280 |
| | MIMIC-TPW2 | 0.84 | −4.69 | 7.16 | 76.88 | 19280 |
| GNSS (2017, Central Himalayas) | Fused TPW | 0.92 | −1.83 | 3.37 | 23.05 | 16623 |
| | ERA5 TPW | 0.91 | −3.28 | 4.63 | 31.71 | 16623 |
| | MIMIC-TPW2 | 0.86 | −3.77 | 4.96 | 33.98 | 16623 |

For the IGRA validation, ERA5 exhibits the highest correlation of 0.98, followed by the fused TPW at 0.96 and MIMIC-TPW2 at 0.89. In terms of accuracy, ERA5 also achieves the smallest RMSE of 1.55 mm and the lowest RRMSE of 19.93 %, while the fused TPW shows slightly larger values of 2.15 mm and 27.66 %, and MIMIC-TPW2 shows the largest error of 3.26 mm and 41.93 %. The higher accuracy of ERA5 in this comparison may be partly attributed to its assimilation of radiosonde observations from the Global Telecommunication System (GTS) network, which includes the IGRA stations

(Durre et al., 2006; Hersbach et al., 2020). The biases are 0.08 mm for the fused TPW, −0.34 mm for ERA5, and −0.38 mm for MIMIC-TPW2, indicating that the fused product has the smallest absolute bias among the three datasets. For the MWR validation, the correlation coefficients are 0.86 for the fused TPW, 0.92 for ERA5, and 0.84 for MIMIC-TPW2. The RMSEs are 6.21 mm, 6.34 mm, and 7.16 mm for ERA5, the fused TPW, and MIMIC-TPW2, respectively, with corresponding RRMSEs of 66.63 %, 68.06 %, and 76.88 %. All three datasets show a clear dry bias, −4.35 mm for ERA5, −3.63 mm for

the fused TPW, and −4.69 mm for MIMIC-TPW2. For the GNSS observations in the central Himalayas, the fused TPW shows a correlation of 0.92, higher than ERA5 at 0.91 and MIMIC-TPW2 at 0.86. Its RMSE (3.37 mm) and relative RMSE (23.05 %) are the smallest among the three datasets, compared with 4.63 mm (31.71 %) for ERA5 and 4.96 mm (33.98 %) for MIMIC-TPW2, indicating that the fused product achieves the highest accuracy in this region. The biases are consistently dry across all datasets, with the fused TPW showing the smallest magnitude at −1.83 mm, compared with −3.28 mm for ERA5 and −3.77 mm for MIMIC-TPW2.

According to these validation results, the accuracy is generally higher when evaluated using IGRA data, but becomes lower when using the scientific expedition data such as MWR and GNSS from the central Himalayas. Compared with global public datasets like IGRA, these expedition observations are located in more remote regions of the Plateau and are crucial for analyzing and validating water vapor transport across the TP. However, all datasets still show relatively large errors in such areas, and further improvements in fusion accuracy are needed for complex and data-sparse regions.

### 4.1.4 Validation of the fused TPW dataset for all years

To comprehensively evaluate the reliability of the fused TPW dataset, statistical validation was performed against GNSS observations from 2016 to 2022 under three weather conditions: all-weather, clear-sky, and cloudy-sky. The results are presented in Tab. 4 (all-weather), Tab. 5 (clear-sky), and Tab. 6 (cloudy-sky).

**Table 4. Statistical validation of the fused TPW dataset against GNSS observations under all-weather conditions for all years (2016–2022).**

| Year | R | Bias (mm) | RMSE (mm) | RRMSE (%) | N |
|------|------|-----------|-----------|-----------|--------|
| 2016 | 0.91 | -1.49 | 3.98 | 43.09 | 253194 |
| 2017 | 0.91 | -1.39 | 3.82 | 40.38 | 285733 |
| 2018 | 0.92 | -1.23 | 3.82 | 40.43 | 311143 |
| 2019 | 0.91 | -1.02 | 3.37 | 39.99 | 174298 |
| 2020 | 0.91 | -1.39 | 4.24 | 38.68 | 201789 |
| 2021 | 0.92 | -0.99 | 3.50 | 38.69 | 300686 |
| 2022 | 0.92 | -1.04 | 3.56 | 39.37 | 279514 |

**Table 5. Statistical validation of the fused TPW dataset against GNSS observations under clear-sky conditions for all years (2016–2022).**

| Year | R | Bias (mm) | RMSE (mm) | RRMSE (%) | N |
|---|---|---|---|---|---|
| 2016 | 0.92 | -0.06 | 2.41 | 35.40 | 124831 |
| 2017 | 0.95 | 0.38 | 1.94 | 27.24 | 143670 |
| 2018 | 0.95 | 0.27 | 2.00 | 27.37 | 142859 |
| 2019 | 0.95 | 0.30 | 1.85 | 27.74 | 78360 |
| 2020 | 0.95 | 0.28 | 2.00 | 24.72 | 106901 |
| 2021 | 0.96 | 0.43 | 1.82 | 26.81 | 138246 |
| 2022 | 0.96 | 0.30 | 1.80 | 26.52 | 136784 |

**Table 6. Statistical validation of the fused TPW dataset against GNSS observations under cloudy-sky conditions for all years (2016–2022).**

| Year | R | Bias (mm) | RMSE (mm) | RRMSE (%) | N |
|---|---|---|---|---|---|
| 2016 | 0.91 | -2.74 | 4.95 | 43.51 | 140840 |
| 2017 | 0.95 | -2.78 | 4.92 | 42.32 | 142063 |
| 2018 | 0.92 | -2.50 | 4.85 | 43.10 | 168284 |
| 2019 | 0.92 | -2.10 | 4.23 | 42.81 | 95938 |
| 2020 | 0.90 | -3.28 | 5.81 | 40.84 | 94988 |
| 2021 | 0.92 | -2.20 | 4.46 | 40.59 | 162440 |
| 2022 | 0.92 | -2.33 | 4.66 | 41.63 | 142730 |

Under all-weather conditions (Tab. 4), the fused TPW dataset exhibited stable performance throughout the study period, with correlation coefficients remaining within 0.91–0.92, RMSE values within 3.37 and 4.24 mm, and slightly dry biases around −0.99 to −1.49 mm. Note that fewer validation samples were available in 2019 and 2020 due to reduced GNSS observations, rather than changes in the fused dataset. Under clear-sky conditions (Tab. 5), the correlation coefficient ranged from 0.92 to 0.96. The Bias was between -0.06 mm and 0.43 mm. The RMSE was from 1.80 mm to 2.41 mm, and the RRMSE ranged from 24.72% to 35.40%. The accuracy under clear-sky conditions is higher than that under all-weather conditions. Under cloudy-sky conditions (Tab. 6), the correlation coefficient ranged from 0.90 to 0.95. The RMSE ranged from 4.23 mm to 5.81 mm, with a relatively higher value in 2020. However, the RRMSE in 2020 was 40.84%, which fell within the fluctuation range of 40.59% to 43.51% observed in other years without abnormal deviation. Overall, the fused TPW dataset showed relatively stable accuracy across different years and weather conditions.

## 4.2 Spatiotemporal distribution analysis of the fused TPW

To evaluate the capability of the fused TPW data to capture the spatiotemporal distribution characteristics of water vapor over the TP, a comparative analysis was conducted using hourly TPW fields from the fused TPW, ERA5 TPW, and MIMIC-TPW2 datasets from 05:00 to 11:00 UTC on 8 June 2017. As shown in Fig. 7, the first column shows the fused TPW, the second column shows ERA5 TPW, and the third column shows MIMIC-TPW2. The three datasets exhibited generally consistent spatial patterns. The TPW in the area west of the Qiangtang Plateau (Area A) is relatively low, while in the northeastern Tarim Basin (Area B), the eastern Sanjiangyuan region of the plateau (Area C), the southeastern Yarlung Zsangbo Grand Canyon (Area D), and the southern edge of the Himalayas (Area E), the TPW is relatively abundant. However, the fused TPW shows a finer depiction of water vapor transport processes, clearly capturing localized moisture enhancement and transport pathways. For instance, in the Sanjiangyuan region (Region C), abundant water vapor is gradually transported northward from the southeast, resulting in accumulation and intensification over time. In contrast, the spatial resolution of ERA5 TPW is insufficient to resolve fine-scale water vapor patterns over complex terrain, leading to overly smooth distributions and weak representation of localized variations. Similarly, MIMIC-TPW2 has coarse spatial resolution and provides a less clear depiction of moisture transport. Moreover, it exhibits noticeable noise in some areas, with abnormal over- and underestimations particularly evident in the eastern part of the TP.

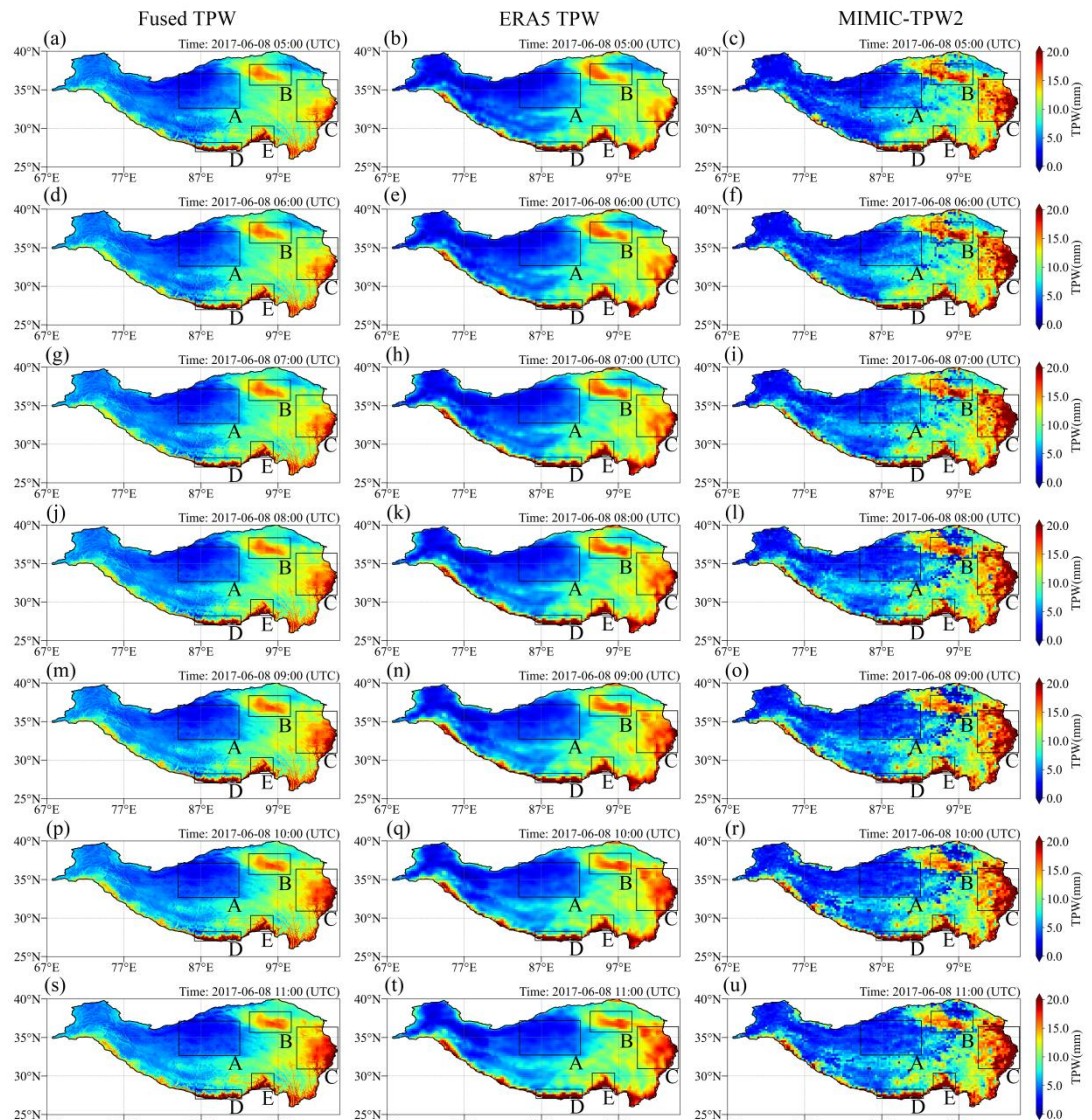

**Figure 7. Spatiotemporal comparison of fused TPW, ERA5 TPW, and MIMIC-TPW2 from 05:00 to 11:00 UTC on 8 June 2017 (first column: fused TPW; second column: ERA5 TPW; third column: MIMIC-TPW2).**

To further evaluate the monthly stability of the fused TPW data, a comparative analysis was conducted between the fused TPW, ERA5 TPW, and MIMIC-TPW2 datasets at the monthly scale. Fig. 8((a), (d), (g), (j)) respectively show the monthly average spatial distribution of the fused TPW in March, June, September, and December 2017, combined with ERA5 TPW (Fig. 8(b), (e), (h), (k)) and MIMIC-TPW2 (Fig. 8(c), (f), (i), (l)). In terms of overall spatial distribution, the three datasets exhibit similar spatial distribution characteristics of water vapor over the TP in different months, with relatively low TPW

values in the plateau's interior and higher TPW values along water vapor transport pathways such as the Tarim Basin and the

southeastern region of the plateau. The fused TPW clearly depicts the spatial gradient of water vapor each month, particularly in complex terrain areas like the Himalayas and Hengduan Mountains, where local water vapor variations are well captured. While the large-scale distribution of ERA5 TPW aligns with that of the fused TPW, its lower spatial resolution results in smoother representations of water vapor gradients. In most regions, the water vapor distribution of MIMIC-TPW2 is comparable to the other two datasets; however, in lake areas such as Qinghai Lake (A), Nam Co, and Serling Co (B), MIMIC-TPW2 shows TPW higher than the surrounding TPW.

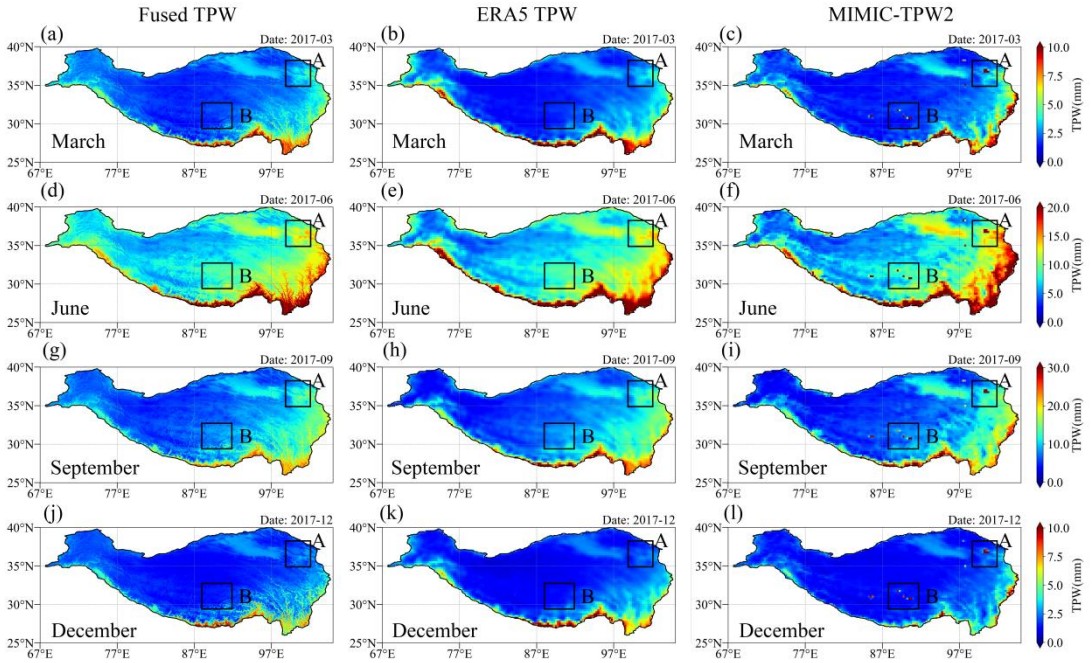

**Figure 8. Comparison of monthly mean spatial distributions of fused TPW, ERA5 TPW, and MIMIC-TPW2 for March, June, September, and December 2017 (first column: fused TPW; second column: ERA5 TPW; third column: MIMIC-TPW2).**

In the above-mentioned monthly-scale spatial distribution comparison results, the TPW values of MIMIC-TPW2 over lake areas such as Qinghai Lake may be overestimated. To verify this phenomenon, the GNSS station QHGC located near Qinghai Lake was selected to provide reference TPW data. The hourly TPW data of MIMIC-TPW2, ERA5 TPW, and the fused TPW during March 21–27, June 11–17, September 1–7, and December 1–7 of 2017 were extracted and compared with the GNSS TPW data, as shown in Fig. 9. It is evident that MIMIC-TPW2 consistently shows higher TPW values than the other datasets during several periods, particularly in summer. For example, in June (Fig. 9(b)), MIMIC-TPW2 exceeds 30 mm on June 12 and 16, while the corresponding GNSS TPW remains below 20 mm. Similar overestimations by MIMIC-TPW2 are also observed in March, September, and December. Compared to MIMIC-TPW2, the fused TPW is more consistent with GNSS TPW, indicating its ability to effectively capture TPW variations. While both the fused TPW and MIMIC-TPW2 rely on similar microwave-based observations, the new fusion algorithm in this study additionally

incorporates high-accuracy TPW from geostationary satellites and applies a bias correction strategy to reduce errors in areas such as lakes.

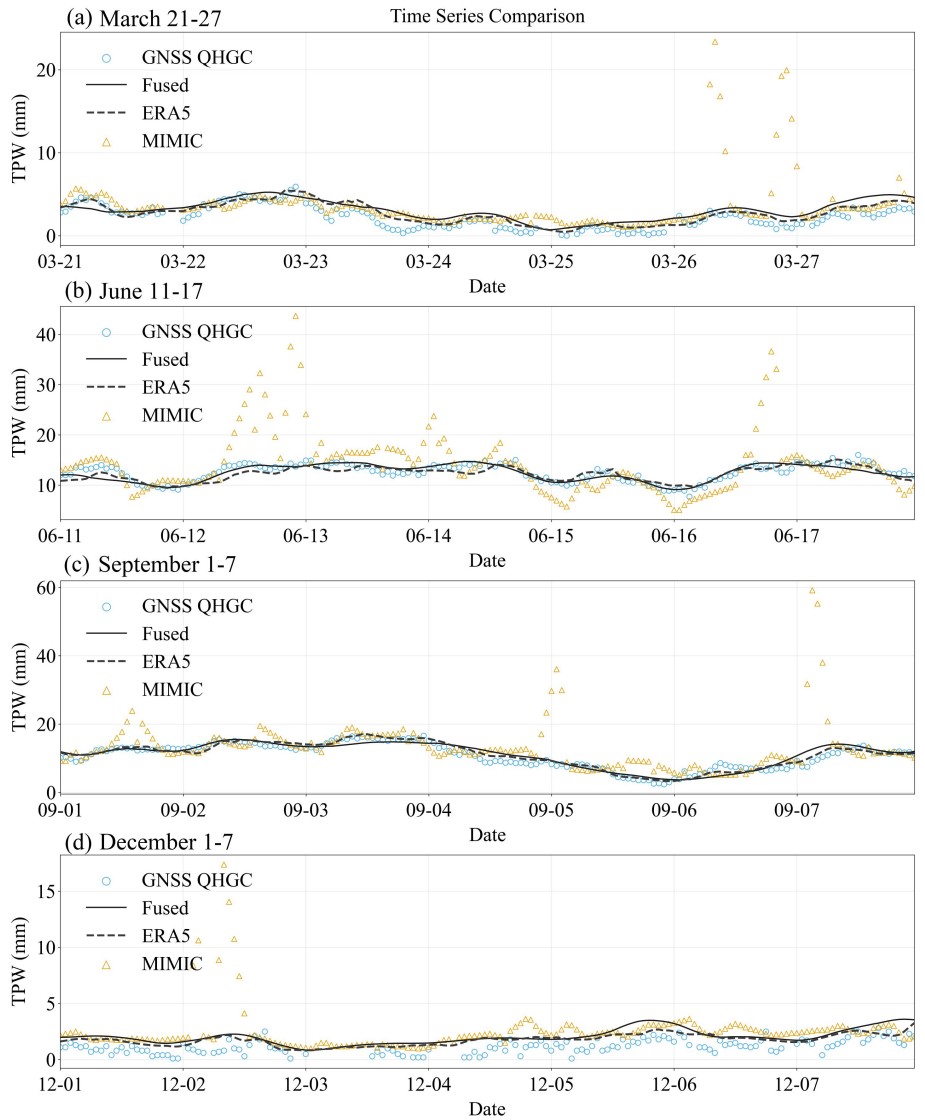

**Figure 9. Hourly time series comparison of fused TPW, ERA5 TPW, and MIMIC-TPW2 at the QHGC GNSS station during different periods in 2017: (a) March 21–27; (b) June 11–17; (c) September 1–7; (d) December 1–7.**

**4.3 Case study**

The meridional atmospheric water vapor transport across the Yarlung Zsangbo Grand Canyon has a significant impact on the precipitation in the southeastern part of the TP (Chen et al., 2024). However, under the complex terrain conditions of the TP, the water vapor transport process shows obvious local characteristics and is difficult to accurately identify through low-

resolution data. The fused TPW product, with its high spatiotemporal resolution and all-weather coverage, is expected to

improve the depiction of such moisture transport processes. To evaluate its practical performance, we analyzed water vapor

transport over the upper Yarlung Zsangbo River and surrounding areas (Fig. 10(a)-(b)) from 09:00 to 18:00 on 8 June 2017,

based on the fused TPW data. In addition, we compared the fused TPW with ERA5 TPW and MIMIC-TPW2 to assess their

capabilities in resolving regional moisture distributions and transient transport features over short time intervals.

Fig. 10(c)-(f) show the fused TPW, which clearly illustrates an east-to-west transport of water vapor forming a continuous

transport corridor along the Yarlung Zsangbo River. The ERA5 TPW (Fig. 10(g)-(j)) show a broadly similar distribution

pattern to the fused TPW, but due to its coarser spatial resolution, the transport features appear blurred, with smoothed

moisture gradients that fail to capture localized transport processes. The MIMIC-TPW2 data (Fig. 10(k)-(n)) exhibit more

unstable water vapor distributions, with pronounced gradient fluctuations in localized areas and less clearly defined transport

pathways. In contrast, the fused TPW, enhanced by multi-source bias correction and by adaptive calibration for cloudy

conditions, provides high-resolution and spatially continuous water vapor fields, enabling the capture of water vapor

transport processes over complex terrain.

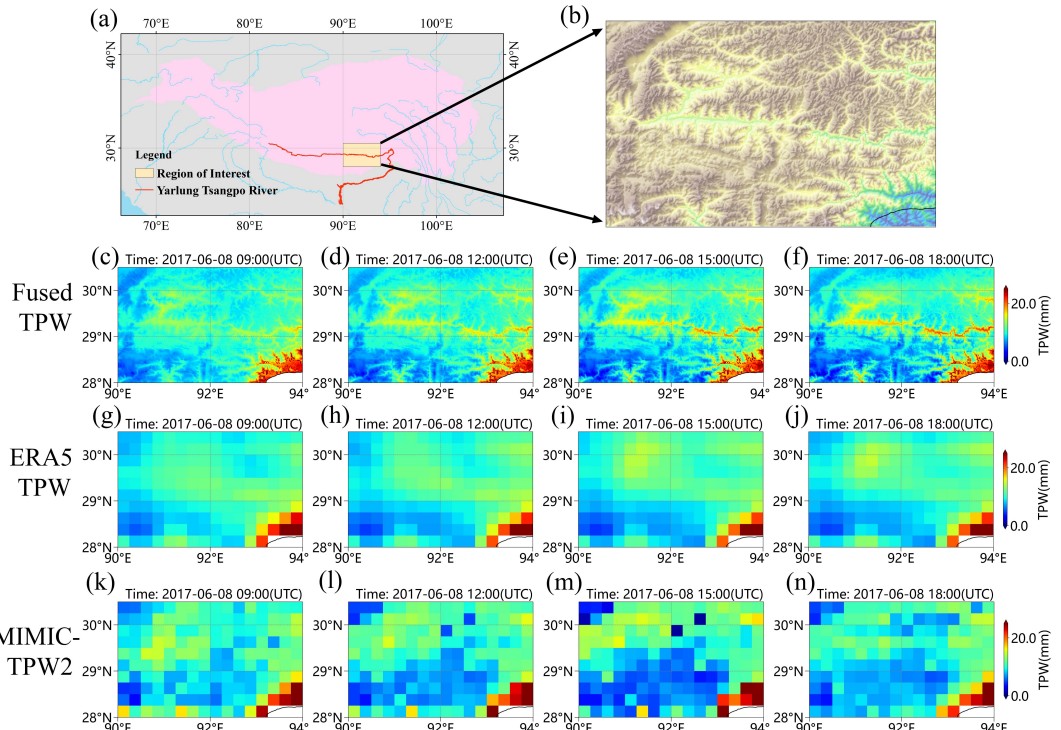

**Figure 10. Hourly spatial distributions of fused TPW, ERA5 TPW, and MIMIC-TPW2 over the upper Yarlung Zsangbo River and surrounding areas. (a: regional location; b: terrain context generated from DEM; c - f: fused TPW; g - j: ERA5 TPW; k - n:**
**MIMIC-TPW2).**

## 5 Discussion

### 5.1 Technical innovations of the TPW fusion framework

Most existing TPW fusion methods rely on ground-based GNSS stations or reanalysis data as references, using spatiotemporal interpolation or machine learning to improve resolution and continuity. However, in regions like the TP, where ground stations are sparse and terrain is complex, these methods face limitations such as insufficient data coverage, low resolution, and persistent systematic errors, which restrict their accuracy in capturing local vapor transport and intense precipitation processes. This study is the first to propose a high spatiotemporal resolution multi-source remote sensing TPW fusion framework for the TP based entirely on satellite observations, combining data from eight microwave remote sensing satellites and H8/9 geostationary satellite. The fusion framework exploits the complementary strengths of different sensors: the all-weather capability of microwave sensors and the high-resolution, high-frequency observations from H8/9. It comprises sequential modules for multi-source bias correction, spatial downscaling, and adaptive correction under cloudy conditions. Using this framework, all-weather TPW dataset covering the TP were generated, with hourly temporal and 0.02° spatial resolution. Technically, the algorithm addresses two major challenges: (1) systematic differences among microwave sensors and (2) the accuracy degradation caused by applying clear-sky-based models in cloudy conditions. Both issues were addressed through targeted correction strategies. To overcome systematic discrepancies among microwave TPW datasets, we developed a satellite-specific bias correction method using H8/9 TPW as a high-accuracy reference. At the coarse-resolution scale, individual correction models were trained for each sensor based on spatiotemporally matched observations, which significantly reduced bias of inter-sensors. These corrected TPW was mosaicked at spatial resolution of 0.25° and was further downscaled to generate an hourly TPW with a spatial resolution of 0.02°. Moreover, in the absence of reliable high-resolution references under cloudy conditions, applying clear-sky-based correction and downscaling models often introduces bias. To address this problem, we proposed a novel adaptive correction scheme based on the assumption of spatiotemporal continuity in the vapor field. Discrepancies between H8/9 TPW and the downscaled TPW along clear-sky boundaries were used to iteratively adjust TPW values under cloudy conditions. This step significantly improves the spatial continuity and accuracy of TPW under cloudy conditions.

### 5.2 Limitations and future improvements of the fusion algorithm

Although the proposed fusion algorithm successfully reconstructs continuous all-weather TPW fields with high temporal and spatial resolution over the Tibetan Plateau, several limitations remain to be addressed.

First, in the western part of the Tibetan Plateau, the original satellite observations show extensive data gaps because of the limited geostationary coverage and the sparse overpasses of microwave sensors. As a result, the available information for spatiotemporal interpolation is relatively insufficient, which may lead to reduced reconstruction accuracy in these regions. Future work will incorporate additional satellite observations—such as data from Fengyun-4A/B (FY-4A/B) and other

geostationary missions—to improve the spatial and temporal coverage of the input datasets, thereby enhancing the quality of reconstructed TPW fields in regions with limited observations.

Second, in regions with persistent and extensive cloud cover, the adaptive correction may experience reduced effectiveness due to the scarcity of valid clear-sky reference pixels from Himawari-8/9 (H8/9). This can lead to locally increased uncertainties, particularly in areas with frequent deep convective systems such as the southern Plateau. Future improvements will focus on introducing additional physical constraints, including cloud microphysical parameters retrieved from infrared or microwave cloud products, and assimilating short-term numerical weather prediction (NWP) fields to enhance correction robustness and continuity under prolonged cloudy conditions.

Third, the current algorithm has been optimized for the Tibetan Plateau, focusing on high-resolution water vapor reconstruction to support regional atmospheric and hydrological studies. Future development can extend this framework toward a near-global scale by integrating multi-geostationary satellite observations to achieve hourly TPW coverage across most low- and mid-latitude regions. For high-latitude areas where geostationary satellites lack coverage, clear-sky TPW retrievals from polar-orbiting optical sensors such as MODIS and MERSI can be incorporated as complementary sources.

In the long term, developing a globally consistent, high-resolution, and purely satellite-based TPW fusion framework will establish a solid observational foundation for quantitative studies of atmospheric moisture transport, energy balance, and land–atmosphere coupling, and will further support the refinement of precipitation (Cui et al., 2025; Ji et al., 2025a), cloud property (Tana et al., 2023, 2025), and radiation estimation (Letu et al., 2023) algorithms across multiple spatial and temporal scales.

### 5.3 Advantages and application prospects of the high-resolution fused TPW dataset

Based on the newly developed fusion algorithm, this study produced an all-weather TPW dataset over the TP from 2016 to 2022, with hourly temporal resolution and 0.02° spatial resolution. Compared with other existing products, this dataset demonstrates advantages in spatial resolution under cloudy conditions, spatiotemporal continuity, and accuracy. Compared with infrared-based TPW retrievals from H8/9 geostationary satellite, this dataset incorporates all-weather TPW data from multiple microwave satellites, thereby addressing the lack of TPW data under cloudy conditions. Compared with products that fuse multi-source microwave TPW data such as MIMIC-TPW2, the new dataset produced in this study has effectively reduced abnormal values over lake areas, along with improved accuracy and spatial continuity. Compared with ERA5 reanalysis data, this fused data has improved the spatial resolution by approximately 12.5 times, making it more suitable for describing the detailed structure of atmospheric water vapor under the complex terrain conditions of the TP, including small-scale transport paths, water vapor gradients, and local accumulation processes. In summary, this dataset is more suitable for the demand for high spatiotemporal resolution water vapor observations over the TP in terms of spatiotemporal coverage and accuracy characteristics than other datasets. It can be used for the calculation of water vapor flux and its divergence, water

vapor budget analysis, identification of precursors to heavy precipitation, providing initial field information for regional numerical models, and improving regional water cycle research and water resource change monitoring (Yao et al., 2019).

**6 Data availability**

The new TPW dataset is published by the National Tibetan Plateau Data Center and is available at: https://doi.org/10.11888/Atmos.tpdc.301518 (Ji et al., 2025b).

**7 Conclusion**

This study proposed a novel multi-source remote sensing data fusion framework based on the concept of a virtual satellite
constellation, which integrated TPW under all-weather conditions from eight microwave remote sensing satellites (DMSP-F17/F18, NOAA-18/19, MetOp-A/B, GCOM-W1, and GPM Core) and TPW with high-temporal-resolution under clear-sky conditions from the H8/9 geostationary satellites. The framework fully exploits the all-weather capability of microwave remote sensing observations and the stable, high resolution infrared observations from geostationary satellites. It also develops key techniques for addressing challenges in complex terrain and cloudy conditions over the TP, including multi-
source data synchronization, systematic bias correction for multi-source TPW, spatial downscaling, and adaptive calibration under cloudy conditions. Based on the proposed algorithm, an all-weather TPW dataset was generated for the TP, covering the period from 2016 to 2022, with a spatial resolution of 0.02° and an hourly temporal resolution. The dataset has been publicly released through the National Tibetan Plateau Data Center (https://doi.org/10.11888/Atmos.tpdc.301518, Ji et al., 2025b).
To evaluate the accuracy of the fused TPW product, this study used the fused TPW data in 2017 was validated against GNSS TPW data under all-weather conditions. As a comparison, the ERA5 TPW and MIMIC-TPW2 were also validated against GNSS TPW. At hourly, daily, and monthly time scales, the RMSEs of the fused TPW are 3.79 mm, 3.50 mm and 3.19 mm respectively. Compared with MIMIC-TPW2, the errors are reduced by 10.82%, 10.26%, 12.84%, and compared with ERA5, they are reduced by 6.19%, 9.60%, 15.16%. In topographically complex and station-sparse regions such as the Himalayas,
the fused TPW shows lower errors at station scale. In terms of spatial distribution and water vapor transport representation, the fused TPW, after bias correction, effectively reduced abnormal values over lake regions such as Qinghai Lake compared to MIMIC-TPW2. Compared with ERA5 TPW, the spatial resolution of the fused TPW is improved by approximately 12.5 times, enabling a clearer depiction of water vapor gradients and transport pathways in complex terrain. During episodes of enhanced moisture transport, the fused TPW effectively captured local moisture buildup in the Yarlung Zsangbo River
Canyon and clearly depicted the associated vapor transport pathways.

The high-resolution TPW dataset developed in this study can be applied to the estimation of water vapor flux, water balance analysis, heavy precipitation forecasting, and regional water resource monitoring over the TP. In the future, by incorporating

multi-geostationary satellite network observations, the proposed algorithm can be extended to global hourly TPW reconstruction at kilometer-scale resolution, providing essential data support for the refined monitoring of large-scale weather systems, the analysis of convective system evolution, and global water cycle research.

**Author contributions**

QS conceived and implemented the fusion framework, developed the methodology, processed the data, performed the validation and analysis, and wrote the original draft. DJ and HL supervised the study, proposed the core ideas, revised the manuscript, and contributed to the design of correction strategies and data integration. HL also provided project coordination and funding support. YW provided key datasets and assisted in data preparation. PZ and HLg supported the study through funding acquisition and project management. CS and SY offered scientific advice and manuscript feedback. JS provided strategic support, project oversight, and research resources.

**Acknowledgments**

We sincerely thank the data providers that supported this study. The microwave TPW products from the Microwave Integrated Retrieval System (MIRS) were obtained via NOAA CLASS (https://www.class.noaa.gov), and the Himawari-8/9 TPW data were provided by the National Tibetan Plateau / Third Pole Environment Data Center (https://data.tpdc.ac.cn). The MIMIC-TPW2 dataset was provided by the Cooperative Institute for Meteorological Satellite Studies (CIMSS) at the University of Wisconsin–Madison (https://tropic.ssec.wisc.edu). ERA5 TPW data were downloaded from the Copernicus Climate Data Store (https://cds.climate.copernicus.eu), and the elevation data were sourced from NOAA NCEI (https://www.ncei.noaa.gov/). We are grateful to Dr. Hongxing Zhang for providing the ground-based GNSS TPW data used for validation, Dr. Xuelong Chen for providing the ground-based microwave radiometer (MWR) observations, and Prof. Kun Yang for sharing the GNSS TPW data from the central Himalayas. We also thank all members of the Second Tibetan Plateau Scientific Expedition for their efforts in establishing and maintaining the observation networks used in this study.

**Financial support**

This work was supported by the National Natural Science Foundation of China (Grant No. 42025504), the National Natural Science Foundation of China (Grant No. U2442214), the National Key Research and Development Program of China (Grant No. 2023YFB3907701), the Second Tibetan Plateau Scientific Expedition and Research Program (Grant No. 2019QZKK0206), and the project "Theory, methods, and experimental validation of electromagnetic and geosphere interactions – Subtask I" (Grant No. E4Z202021F).

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
