# Peer review of "An hourly $0.02^{\circ}$ total precipitable water dataset for all-weather conditions over the Tibetan Plateau through the fusion of observations of geostationary and multi-source microwave satellites"

_Earth System Science Data, 2025_

## Referee Comment (RC1)

**Comments to the Author,**

This manuscript integrates TPW products from eight microwave satellites with those from the Himawari-8/9 (H8/9) geostationary satellite to generate an all-weather TPW dataset at the highest spatiotemporal resolution (0.02°). Methodologically, two correction strategies are developed. First, a bias correction approach is introduced, using H8/9 TPW data as a reference to calibrate multi-source microwave remote sensing TPW and thereby reduce inter-sensor discrepancies. Second, an adaptive correction method is designed to improve the accuracy and spatial continuity of the fused TPW data under cloudy conditions. Compared with ERA5, which has a spatial resolution of 0.25°, the fused product represents a 12.5-fold improvement in spatial resolution. Overall, this is an interesting topic and the methodology is appropriate. However, I find several issues that require further clarification, as outlined below. I recommend the manuscript needs revision before accepting for publication.

**Major comments:**

- 1. Lines 361–363:The correction model is trained using a random forest algorithm. Please clarify which year of data was used for training and how many samples were included. Furthermore, how is the model trained, updated, and applied on a monthly basis?
- 2. Similarly, for the downscaling of coarse-resolution TPW, from which time period are the datasets used to train the spatial downscaling model and the correction model, and how many samples are included?
- 3. Line 371: The authors state that "for the remaining missing areas, linear interpolation was used to fill the data pixel by pixel." As shown in Fig. 4c, in some regions—for example, over the Tibetan Plateau—TPW data are available only in the central and eastern parts, while large areas in the west remain missing. In such cases, how is interpolation performed to generate the fused TPW data at 0.25° resolution?
- 4. The fused TPW data under all-weather conditions include clear-sky conditions derived from the H8/9 TPW dataset, which is retrieved using a neural network-based rapid retrieval algorithm (Jiang et al., 2022). Whereas the high-resolution TPW data under cloudy conditions, derived from eight microwave satellites, are used to fill the missing areas in the H8/9 TPW dataset. Therefore, it is recommended that the authors provide an evaluation of the fused TPW specifically under cloudy conditions to assess its accuracy and reliability.
- 5. The authors have constructed an all-weather TPW dataset with hourly temporal and  $0.02^{\circ}$  spatial resolution covering the Tibetan Plateau from 2016 to 2022. However, the accuracy evaluation is performed only for the TPW in 2017, and no assessment in the other years is given in the current manuscript. It is recommended that the authors evaluate the TPW data for all years from 2016 to 2022 to ensure the robustness and consistency of the dataset.
- 6. The fused high-resolution TPW product can be compared with radiosonde-derived TPW to further validate its accuracy, as radiosonde measurements are considered a reliable method for obtaining TPW with minimal uncertainty. As shown in line 230, the manuscript mentions a comparison with TPW data from radiosonde observations (IGRA). Here, it unclear how well the

fused high-resolution TPW product agrees with the IGRA data. A quantitative assessment or validation would help clarify this point.

**Other minor comments:**

- 1. Line 218: The hyphen in "3.5 5.2 mm" should be replaced with an en dash, i.e., "3.5-5.2 mm," to conform to standard scientific notation.
- 2.The sub-subsection titles formats need to be standardized. For example, line 254 is titled "2.2.3.GNSS TPW data," whereas line 276 is titled "2.2.5ERA5 TPW data." It is recommended to include a consistent separator, such as a period or space, between the subsection number and the title to avoid potential confusion for readers.
- 3. Line 287, the term "spatiotemporal information" is used. Does this refer to the "time" mentioned in line 292?
- 4. In Figure 2, within the bias correction section, the coarse-resolution TPW panel for 2017-06-08 10:00 (UTC) appears to have an extra arrow in the lower-right corner (around 67°E–77°E). Please check this.
- 5.In Figure 3, the maximum difference in scanning times among the satellites is approximately 3 hours, while the minimum difference is about 1 hour. Does this imply that the temporal resolution of the fused TPW data in this study ranges from 1 to 3 hours?
- 6.Line 357: The authors mention using the random forest algorithm. A proper reference for the random forest method should be provided to support its use.
- 7.Lines 459–460: The manuscript states that "the fused TPW product generated for the same year was validated under all-weather conditions at three temporal scales: hourly, daily, and monthly." Could the authors clarify how the fused TPW data are processed at different temporal scales? For example, do "daily" and "monthly" refer to daily and monthly mean TPW, respectively? Additionally, was the evaluation conducted using the entire year of 2017?
- 8. Line 483: "figure" should be "Figure 6".
- 9. The citation style for figures should be kept consistent throughout the manuscript. For example, line 481 cites "...(Fig. 6a, c, e)," whereas line 513 cites "...(Fig. 8(c, f, i, l))." Please adopt a uniform format for figure references.

---

## Author Comment (AC1)

**Response to RC1**

Overall, this is an interesting topic and the methodology is appropriate. However, I find several issues that require further clarification, as outlined below. I recommend the manuscript needs revision before accepting for publication.

**Response:** We sincerely thank you for the constructive and insightful comments. Each comment has been carefully considered, and the corresponding revisions have been incorporated into the revised manuscript. Our detailed point-by-point responses are provided below.

**1. Major comments**

**Comment 1-1:** Lines 361–363: The correction model is trained using a random forest algorithm. Please clarify which year of data was used for training and how many samples were included. Furthermore, how is the model trained, updated, and applied on a monthly basis?

Response: Thank you for this suggestion. We have added clarifications in the revised manuscript regarding the data period, the sample size, and the update mechanism of the correction model. Specifically, for each microwave satellite, the bias correction model was independently trained using clear-sky samples of H8/9 TPW and the corresponding microwave TPW data from each month of each year. For instance, in January 2017, 11565 valid collocations between NOAA-19 and H8 were used to train the monthly correction model. This model construction was repeated for each satellite and each month, allowing the bias correction to be updated both temporally (month by month) and across sensors. The model was built under clear-sky conditions and then applied to all-weather situations to ensure the consistency of bias adjustment. The corresponding clarification has been added in Section 3.2 (Lines 380–384) of the revised manuscript. The bold sentences below indicate the revised content:

"To address the temporal and algorithmic discrepancies among multi-source microwave data, this study employs the random forest algorithm (Breiman, 2001) to develop a correction model. The target data is H8/9 TPW data under clear-sky conditions, and the model training input data include microwave remote sensing TPW data, longitude, latitude, elevation, and

time, etc. The model was trained and updated separately for each satellite and for each month of every year to account for seasonal variations. For example, the correction model for NOAA-19 in January 2017 was trained using 11,565 valid collocated clear-sky samples with H8 TPW. The correction model was constructed under clear-sky conditions and applied to all-weather conditions. This month-by-month, satellite-specific training ensures that systematic bias correction was dynamically adapted to both temporal and sensor-related variations."

**Comment 1-2**: Similarly, for the downscaling of coarse-resolution TPW, from which time period are the datasets used to train the spatial downscaling model and the correction model, and how many samples are included?

Response: Thank you for the question. We have added the details regarding the data period, sample size, and model updating procedure for the downscaling process. The downscaling consists of two sequential steps: (1) a DEM-based spatial downscaling model and (2) a refinement model using high-resolution H8/9 observations. Both models were trained using data within ±3 hours of the target time, following the temporal sampling strategy adopted in Sun et al. (2024). For example, on 1 January 2017, the DEM-based downscaling model used 27,360 valid samples, while the H8-based correction model used 487,986 valid samples. The DEM-based downscaling model was trained using input and output data at the 0.25° resolution in all-weather conditions, and subsequently applied to input variables at 0.02° resolution to produce preliminary hourly high-resolution TPW data. The H8-based correction model was trained and updated for each hour under clear-sky conditions and then applied to all-weather conditions. This information has been added to the revised manuscript (Lines 424-428, Lines 439-440). The bold sentences below indicate the revised content:

"Preliminary spatial downscaling of coarse-resolution reconstructed TPW data. To enhance the spatial resolution of coarse-resolution reconstructed TPW data in regions with significant topographic variation, a spatial downscaling model was developed using the random forest algorithm. The 0.25° coarse-resolution reconstructed TPW data served as the target variable, while elevation, latitude, longitude, and time were used as input features to account for the effects of spatial location, topographic characteristics, and temporal variation on TPW

distribution. The model was trained using input and output data at the 0.25° resolution, and subsequently applied to input variables at 0.02° resolution to produce preliminary hourly high-resolution TPW data on all-weather conditions. Following the temporal sampling strategy proposed by Sun et al. (2024), training samples were selected within a ±3-hour window centered on each target hour. For instance, on 1 January 2017, the DEM-based downscaling model was trained using 27,360 valid samples.

Refining the preliminary downscaling with H8/9 high-resolution TPW data. Although the spatial resolution of the preliminary downscaling results was improved, there might still be spatial detail deviations due to the lack of actual high-resolution water vapor observation information. Therefore, the H8/9 TPW data under clear-sky conditions was further introduced as a reference to correct the preliminary downscaling results. Before constructing the correction model, quality control was conducted on the training data samples. Specifically, the differences between the preliminarily downscaled TPW and H8/9 TPW were calculated, and only the samples with differences within ±3 standard deviations from the mean were retained. Subsequently, with H8/9 TPW as the target variable and the preliminary downscaling TPW as the input variable, combined with auxiliary information such as longitude, latitude, elevation, and time, a random forest algorithm was used to construct the correction model. This model was trained under clear-sky conditions and applied to all-weather conditions to correct the spatial distribution of the preliminary downscaled TPW data, obtaining hourly, 0.02° resolution downscaling TPW data (DS TPW). For instance, on 1 January 2017, the H8-based correction model was trained using 487,986 valid samples."

Reference: Sun, Q., Ji, D., Letu, H., Ni, X., Zhang, H., Wang, Y., Li, B., and Shi, J.: A method for estimating high spatial resolution total precipitable water in all-weather condition by fusing satellite near-infrared and microwave observations, Remote Sens. Environ., 302, 113952, https://doi.org/10.1016/j.rse.2023.113952, 2024.

Comment 1-3: Line 371: The authors state that "for the remaining missing areas, linear interpolation was used to fill the data pixel by pixel." As shown in Fig. 4c, in some regions—for example, over the Tibetan Plateau—TPW data are available only in the central and eastern parts, while large areas in the west remain missing. In such cases, how is

interpolation performed to generate the fused TPW data at 0.25° resolution?

**Response:** Thank you for your comment. In the revised manuscript, we have clarified the interpolation process in detail (Section 3.2, Lines 392–399):

"For the remaining missing regions after mosaicking, a combined temporal–spatial interpolation strategy was adopted to achieve continuous hourly coverage. Specifically, linear interpolation was applied along the time dimension between two adjacent valid observations when the temporal gap was less than or equal to 24 hours, while longer gaps or edge segments remained missing. Missing pixels in each hourly 2D TPW field were then interpolated spatially using a bilinear method, but only when more than 50% of the pixels within the surrounding 1°×1° neighborhood were valid. Pixels lacking sufficient valid neighbors were left unfilled. When residual missing values remained after spatial interpolation, the temporal and spatial interpolation procedures were iteratively repeated until no further gaps were present. Finally, a Savitzky–Golay filter was applied to smooth local variations in the reconstructed field, resulting in hourly, 0.25° resolution fused TPW data."

The same interpolation method was applied uniformly across the Plateau, including the western region. However, due to the limited geostationary coverage and the sparse overpasses of microwave sensors, the original satellite observations exhibit extensive data gaps over western Tibetan Plateau. Consequently, the available information for temporal and spatial interpolation is relatively limited, which may reduce reconstruction accuracy in these areas.

To clarify this issue and outline potential improvements, a new Section 5.2 ("Limitations and future improvements of the fusion algorithm") has been added in the revised manuscript (the former Section 5.2 has been renumbered as Section 5.3). This new section explicitly discusses the reconstruction limitation in western Tibetan Plateau and the planned expansion of the data coverage through the integration of additional satellite observations such as FY-4A/B. The newly added text is as follows:

**"5.2 Limitations and future improvements of the fusion algorithm**

Although the proposed fusion algorithm successfully reconstructs continuous all-weather TPW fields with high temporal and spatial resolution over the Tibetan Plateau, several limitations remain to be addressed.

First, in the western part of the Tibetan Plateau, the original satellite observations show

extensive data gaps because of the limited geostationary coverage and the sparse overpasses of microwave sensors. As a result, the available information for spatiotemporal interpolation is relatively insufficient, which may lead to reduced reconstruction accuracy in these regions. Future work will incorporate additional satellite observations—such as data from Fengyun-4A/B (FY-4A/B) and other geostationary missions—to improve the spatial and temporal coverage of the input datasets, thereby enhancing the quality of reconstructed TPW fields in regions with limited observations.

Second, in regions with persistent and extensive cloud cover, the adaptive correction may experience reduced effectiveness due to the scarcity of valid clear-sky reference pixels from Himawari-8/9 (H8/9). This can lead to locally increased uncertainties, particularly in areas with frequent deep convective systems such as the southern Plateau. Future improvements will focus on introducing additional physical constraints, including cloud microphysical parameters retrieved from infrared or microwave cloud products, and assimilating short-term numerical weather prediction (NWP) fields to enhance correction robustness and continuity under prolonged cloudy conditions.

Third, the current algorithm has been optimized for the Tibetan Plateau, focusing on high-resolution water vapor reconstruction to support regional atmospheric and hydrological studies. Future development can extend this framework toward a near-global scale by integrating multi-geostationary satellite observations to achieve hourly TPW coverage across most low- and mid-latitude regions. For high-latitude areas where geostationary satellites lack coverage, clear-sky TPW retrievals from polar-orbiting optical sensors such as MODIS and MERSI can be incorporated as complementary sources.

In the long term, developing a globally consistent, high-resolution, and purely satellite-based TPW fusion framework will establish a solid observational foundation for quantitative studies of atmospheric moisture transport, energy balance, and land-atmosphere coupling, and will further support the refinement of precipitation (Cui et al., 2025; Ji et al., 2025a), cloud property (Tana et al., 2023, 2025), and radiation estimation (Letu et al., 2023) algorithms across multiple spatial and temporal scales."

Comment 1-4: The fused TPW data under all-weather conditions include clear-sky conditions derived from the H8/9 TPW dataset, which is retrieved using a neural network-based rapid retrieval algorithm (Jiang et al., 2022). Whereas the high-resolution TPW data under cloudy conditions, derived from eight microwave satellites, are used to fill the missing areas in the H8/9 TPW dataset. Therefore, it is recommended that the authors provide an evaluation of the fused TPW specifically under cloudy conditions to assess its accuracy and reliability.

**Response:** Thank you for this suggestion. In the revised manuscript, we have added a new subsection (Section 4.1.1) presenting detailed validation results of the fused TPW under cloudy conditions using 2017 GNSS observations. The evaluation also includes a comparison with the Himawari-8 (H8) TPW and the multi-source microwave (MW) TPW retrievals on different weathers. The corresponding results are summarized in Table 2.

Table 2. Comparison of hourly TPW from H8 TPW, MW TPW, and fused TPW against GNSS observations in 2017.

| Weather    | Data tyma | D    | Diag (mm) | RMSE | RRMSE | N      |  |
|------------|-----------|------|-----------|------|-------|--------|--|
| conditions | Data type | R    | Bias (mm) | (mm) | (%)   |        |  |
| Clear sky  | H8 TPW    | 0.95 | 0.38      | 1.94 | 27.24 | 143670 |  |
|            | MW TPW    | 0.89 | -1.82     | 3.79 | 54.37 | 42367  |  |
| Cloudy     | MW TPW    | 0.88 | -4.31     | 6.44 | 55.44 | 46545  |  |
|            | Fused     |      | 2.79      | 4.92 | 42.32 | 142062 |  |
|            | TPW       | 0.95 | -2.78     |      |       | 142063 |  |

The results show that under clear-sky conditions, the H8 TPW achieves the highest accuracy, while the MW TPW has relatively larger errors. Under cloudy conditions, the fused TPW demonstrates a clear improvement over the MW TPW, with reduced bias and RMSE.

The following text has been added to the revised manuscript (Lines 389–502). The bold sentences below indicate the revised content:

**"4.1.1 Validation of TPW under clear-sky and cloudy conditions**

To evaluate the performance of the fusion algorithm under different sky conditions, the 2017 GNSS TPW observations were used to validate the H8 TPW, the multi-source MW

TPW, and the fused TPW data. During validation, each GNSS station was matched with the nearest remote sensing grid cell, and GNSS measurements at hourly timestamps were used to validate the corresponding fused TPW. The corresponding results are summarized in Tab. 2. Under clear-sky conditions, the H8 TPW shows the highest correlation coefficient of 0.95, followed by 0.89 for the MW TPW. The biases are 0.38 mm for H8 and -1.82 mm for MW. The RMSE values are 1.94 mm for H8 and 3.79 mm for MW, confirming that the H8 TPW product provides the most reliable estimates under clear-sky conditions and justifying its use as the reference for bias correction. Under cloudy conditions, the fused TPW maintains a correlation coefficient of 0.95, higher than 0.88 for the MW TPW. The biases are -2.78 mm for the fused TPW and -4.31 mm for MW, and the RMSE values are 4.92 mm and 6.44 mm, respectively. These results demonstrate that the fusion algorithm effectively reduces bias and error under cloudy conditions, leading to a notable improvement in accuracy compared with the original MW observations.

Table 2. Comparison of hourly TPW from Himawari-8, multi-source microwave (MW), and fused products against GNSS observations in 2017.

| Weather    | Data type | R    | Diag (mm)     | RMSE | RRMSE | N      |
|------------|-----------|------|---------------|------|-------|--------|
| conditions |           |      | Bias (mm)     | (mm) | (%)   |        |
| Clear sky  | H8 TPW    | 0.95 | 0.38          | 1.94 | 27.24 | 143670 |
|            | MW TPW    | 0.89 | -1.82         | 3.79 | 54.37 | 42367  |
| Cloudy     | MW TPW    | 0.88 | -4.31         | 6.44 | 55.44 | 46545  |
|            | Fused     | 0.95 | 4. T 0 | 4.92 | 42.22 |        |
|            | TPW       |      | -2.78         |      | 42.32 | 142063 |

"

Comment 1-5: The authors have constructed an all-weather TPW dataset with hourly temporal and 0.02° spatial resolution covering the Tibetan Plateau from 2016 to 2022. However, the accuracy evaluation is performed only for the TPW in 2017, and no assessment in the other years is given in the current manuscript. It is recommended that the authors

evaluate the TPW data for all years from 2016 to 2022 to ensure the robustness and consistency of the dataset.

**Response:** Thank you for this suggestion. In the revised manuscript, we have extended the validation to cover all years from 2016 to 2022, using hourly GNSS TPW observations to assess the interannual consistency and robustness of the fused TPW dataset. The corresponding results are summarized in the table below.

Table 4. Statistical validation of the fused TPW dataset against GNSS observations for all years (2016–2022).

| Year | R    | Bias (mm) | RMSE (mm) | RRMSE (%) | N      |
|------|------|-----------|-----------|-----------|--------|
| 2016 | 0.91 | -1.49     | 3.98      | 43.09     | 253194 |
| 2017 | 0.91 | -1.39     | 3.82      | 40.38     | 285733 |
| 2018 | 0.92 | -1.23     | 3.82      | 40.43     | 311143 |
| 2019 | 0.91 | -1.02     | 3.37      | 39.99     | 174298 |
| 2020 | 0.91 | -1.39     | 4.24      | 38.68     | 201789 |
| 2021 | 0.92 | -0.99     | 3.50      | 38.69     | 300686 |
| 2022 | 0.92 | -1.04     | 3.56      | 39.37     | 279514 |

The results show consistent performance across all years, with correlation coefficients remaining stable between 0.91–0.92, RMSE values within 3.37–4.24 mm, and slightly dry biases around –0.99 to –1.49 mm. Note that fewer validation samples were available in 2019 and 2020 due to reduced GNSS observations, rather than changes in the fused dataset. Overall, these findings confirm the robust accuracy and temporal stability of the fused TPW product.

The following text has been added in the revised manuscript (Lines 567–576). The bold sentences below indicate the revised content:

**"4.1.4 Validation of the fused TPW dataset for all years**

To further assess the robustness and interannual stability of the fused TPW dataset, the validation was extended to cover all years from 2016 to 2022, using hourly GNSS TPW observations as reference. The corresponding statistical results are summarized in Tab. 4.

Table 4. Statistical validation of the fused TPW dataset against GNSS observations for

all years (2016-2022).

| Year | R    | Bias (mm) | RMSE | RRMSE | N      |
|------|------|-----------|------|-------|--------|
|      |      |           | (mm) | (%)   | N      |
| 2016 | 0.91 | -1.49     | 3.98 | 43.09 | 253194 |
| 2017 | 0.91 | -1.39     | 3.82 | 40.38 | 285733 |
| 2018 | 0.92 | -1.23     | 3.82 | 40.43 | 311143 |
| 2019 | 0.91 | -1.02     | 3.37 | 39.99 | 174298 |
| 2020 | 0.91 | -1.39     | 4.24 | 38.68 | 201789 |
| 2021 | 0.92 | -0.99     | 3.50 | 38.69 | 300686 |
| 2022 | 0.92 | -1.04     | 3.56 | 39.37 | 279514 |

The results show consistent performance across all years, with correlation coefficients remaining stable between 0.91–0.92, RMSE values within 3.37–4.24 mm, and slightly dry biases around -0.99 to -1.49 mm. Note that fewer validation samples were available in 2019 and 2020 due to reduced GNSS observations, rather than changes in the fused dataset. Overall, these findings confirm that the fused TPW dataset maintains reliable accuracy and interannual consistency throughout the 2016–2022 period."

Comment 1-6: The fused high-resolution TPW product can be compared with radiosonde-derived TPW to further validate its accuracy, as radiosonde measurements are considered a reliable method for obtaining TPW with minimal uncertainty. As shown in line 230, the manuscript mentions a comparison with TPW data from radiosonde observations (IGRA). Here, it unclear how well the fused high-resolution TPW product agrees with the IGRA data. A quantitative assessment or validation would help clarify this point.

**Response:** Thank you for the suggestion. In the revised manuscript, we have added the quantitative validation results of the fused TPW product using IGRA TPW data from 2022, and compared them with ERA5 TPW and MIMIC-TPW2 to provide a clear assessment of product accuracy. The corresponding text has been added in the revised manuscript (Lines 539–566). The bold sentences below indicate the revised content:

"To further evaluate the performance of the fused TPW dataset, additional validations

were conducted using (1) IGRA radiosonde TPW data from 2022, (2) microwave radiometer (MWR) observations obtained from the scientific expedition over the Tibetan Plateau in 2022, and (3) GNSS TPW observations from the central Himalayas in 2017. Table.3 shows the metrics comparisons between the fused TPW, ERA5, and MIMIC-TPW2 datasets.

Table 3. Validation of the fused TPW, ERA5 TPW, and MIMIC-TPW2 products using independent observations from IGRA, ground-based MWR, and GNSS over the central Himalayas.

| Observation       | Data Type         | R    | Bias  | RMSE | RRMSE | N     |
|-------------------|-------------------|------|-------|------|-------|-------|
| Туре              | Ба са Туре | K .  | (mm)  | (mm) | (%)   |       |
|                   | Fused TPW         | 0.96 | 0.08  | 2.15 | 27.66 | 5059  |
| IGRA
(2022)    | ERA5 TPW          | 0.98 | -0.34 | 1.55 | 19.93 | 5059  |
|                   | MIMIC-TPW2        | 0.89 | -0.38 | 3.26 | 41.93 | 5059  |
|                   | Fused TPW         | 0.86 | -3.63 | 6.34 | 68.06 | 19280 |
| MWR
(2022)     | ERA5 TPW          | 0.92 | -4.35 | 6.21 | 66.63 | 19280 |
|                   | MIMIC-TPW2        | 0.84 | -4.69 | 7.16 | 76.88 | 19280 |
| GNSS              | Fused TPW         | 0.92 | -1.83 | 3.37 | 23.05 | 16623 |
| (2017,
Central | ERA5 TPW          | 0.91 | -3.28 | 4.63 | 31.71 | 16623 |
| Himalayas)        | MIMIC-TPW2        | 0.86 | -3.77 | 4.96 | 33.98 | 16623 |

For the IGRA validation, ERA5 exhibits the highest correlation of 0.98, followed by the fused TPW at 0.96 and MIMIC-TPW2 at 0.89. In terms of accuracy, ERA5 also achieves the smallest RMSE of 1.55 mm and the lowest RRMSE of 19.93 %, while the fused TPW shows slightly larger values of 2.15 mm and 27.66 %, and MIMIC-TPW2 shows the largest error of 3.26 mm and 41.93 %. The higher accuracy of ERA5 in this comparison may be partly attributed to its assimilation of radiosonde observations from the Global Telecommunication System (GTS) network, which includes the IGRA stations (Durre et

al., 2006; Hersbach et al., 2020). The biases are 0.08 mm for the fused TPW, -0.34 mm for ERA5, and -0.38 mm for MIMIC-TPW2, indicating that the fused product has the smallest absolute bias among the three datasets."

**2. Minor comments:**

Comment 2-1: Line 218: The hyphen in "3.5 - 5.2 mm" should be replaced with an en dash.

**Response:** Thank you for pointing this out. The hyphen has been replaced with an en dash, and the sentence in Line 218 has been revised to: "Verified by Suomi GPS TPW, the root mean square error (RMSE) of AMSR2 TPW data over global land is in the range of 3.5–5.2 mm.".

**Comment 2-2:** The sub-subsection titles formats need to be standardized.

**Response:** Thank you for the suggestion. We have revised all subsection titles to ensure a consistent format. All subsection and sub-subsection titles have been standardized to use the format "n.n.n [space] Title".

**Comment 2-3:** Line 287, the term "spatiotemporal information" is used. Does this refer to the "time" mentioned in line 292?

Response: Thank you for the question. The term "spatiotemporal information" refers to the geographic and temporal variables (longitude, latitude, and time) mentioned later in the paragraph. The original sentence was "The auxiliary data used in this study mainly include the vector boundary of the TP, elevation data, and spatiotemporal information." To improve clarity, it has been revised to "The auxiliary data used in this study mainly include the vector boundary of the TP, elevation data, and geographic coordinates and time." (Line 303-304 in the revised manuscript).

Comment 2-4: In Figure 2, the coarse-resolution TPW panel for 2017-06-08 10:00 (UTC) appears to have an extra arrow in the lower-right corner.

**Response:** Thank you for the reminder. The redundant arrow in Figure 2 has been removed in the revised manuscript. The corrected Figure 2 is shown below:

Figure 2. Flowchart of the multi-source remote sensing TPW fusion algorithm

**Comment 2-5:** In Figure 3, the maximum difference in scanning times among the satellites is approximately 3 hours, while the minimum is about 1 hour. Does this imply that the temporal resolution of the fused TPW data in this study ranges from 1 to 3 hours?

Response: Thank you for the question. The fused TPW dataset developed in this study is not constrained by the 1–3-hour observation intervals of the microwave satellites. Instead, it provides continuous hourly all-weather coverage across the Tibetan Plateau. In the fusion framework, the hourly H8/9 TPW data provide the baseline observations. At time steps without microwave data, the hourly TPW fields are derived from H8/9 and subsequently reconstructed through spatiotemporal interpolation under cloudy conditions. When

multi-source microwave data are available, they are bias-corrected and used to fill the cloudy gaps in the H8/9 retrievals. If missing values remain, the reconstruction step is repeated until full hourly coverage is achieved.

To clarify this process, the following text has been added in Section 3.1 (Lines 353–357) of the revised manuscript. The bold sentences below indicate the revised content:

"Data within 30 minutes before and after each hour were assigned to that hour, resulting in the construction of hourly microwave remote sensing TPW data. Taking the hourly matching results of different satellites over the TP on March 1, 2017, as an example (Fig. 3), the maximum difference in scanning times among the satellites was approximately 3 hours. The spatiotemporal distribution of the multi-source microwave remote sensing TPW data after hourly mosaicking is shown in Fig. 4(a). In contrast, the H8/9 satellite continuously observes the entire disk region at 10-minute intervals. By directly extracting the observation data at the integer hour, hourly H8/9 TPW data can be generated. The spatiotemporal distribution of the processed H8/9 TPW data is shown in Fig. 4(b). In the fusion framework, the hourly H8/9 TPW data provide the baseline observations. At time steps without microwave data, the hourly TPW fields are derived from H8/9 and subsequently reconstructed through spatiotemporal interpolation under cloudy conditions. When multi-source microwave data are available, they are bias-corrected and used to fill the cloudy gaps in the H8/9 retrievals. If missing values remain, the reconstruction step was repeated until full hourly coverage was achieved."

**Comment 2-6:** Line 357: The authors mention using the random forest algorithm. A proper reference should be provided.

**Response:** Thank you for the suggestion. We have added Breiman (2001) as a reference for the random forest method. The corresponding sentence in the revised manuscript (Line 378) has been revised to:

"To address the temporal and algorithmic discrepancies among multi-source microwave data, this study employs the random forest algorithm (Breiman, 2001) to develop a correction model."

Reference added: Breiman, L.: Random forests, Machine Learning, 45, 5-32,

https://doi.org/10.1023/A:1010933404324, 2001.

Comment 2-7: Lines 459–460: Clarify "hourly, daily, and monthly" validation scales.

**Response:** Thank you for the suggestion. The "hourly" scale refers to direct validation of the hourly fused TPW data, while the "daily" and "monthly" scales represent validations based on daily- and monthly-averaged TPW values derived from the hourly product. The corresponding sentence in the revised manuscript (Lines 506–507) has been supplemented with this clarification as follows:

"To evaluate the fusion algorithm, GNSS TPW data from 2017 were selected as reference, and the fused TPW product generated for the same year was validated under all-weather conditions at three temporal scales: hourly, daily, and monthly. The "hourly" scale refers to direct validation of the hourly fused TPW data, while the "daily" and "monthly" scales represent validations based on daily- and monthly-averaged TPW values derived from the hourly product."

Comment 2-8: Line 483: "figure" should be "Figure 6".

**Response:** Thank you for the suggestion. The term "figure" has been corrected to "Figure 6" in the revised manuscript (Line 526).

**Comment 2-9:** The citation style for figures should be consistent.

**Response:** Thank you for the suggestion. We have carefully checked and unified all figure and table citations throughout the manuscript. The citation style has been standardized as follows:

"Figure n" when appearing at the beginning of a sentence;

"Fig. n" for in-text references;

"Fig. n(x)" for subfigures;

"Fig. n(x)-(y)" for consecutive subfigures;

"Fig. n(k), (q), (x)" for nonconsecutive subfigures.

The same formatting rules have been applied to all table citations to ensure consistency throughout the manuscript.

---

## Author Comment (AC2)

**Response to RC2**

This study presents a ML-based multi-satellite fusion framework, successfully generating a high-resolution, all-weather TPW dataset for the Tibetan Plateau, which represents a significant contribution to the field. However, several methodological limitations and potential avenues for improvement warrant discussion.

**Response:** We sincerely thank Reviewer 2 for the valuable suggestions and thoughtful feedback. The manuscript has been thoroughly revised in response to the comments. Specific responses and the corresponding modifications are presented below.

Comment 1: A primary concern is the framework's heavy reliance on the Himawari-8/9 (H8/9) clear-sky TPW product as the foundational reference for both bias correction and spatial downscaling. While this strategy effectively mitigates inter-sensor biases, it inherently transfers the uncertainties and potential systematic errors of the H8/9 retrievals into the final fused product. Furthermore, the adaptive correction method for cloudy conditions, which extrapolates biases from clear-cloudy boundaries, may see its efficacy diminish in regions of extensive, persistent cloud cover where valid H8/9 reference pixels are distant. Future work could enhance robustness by incorporating cloud physical properties (e.g., from microwave sounders) or assimilating short-term numerical weather prediction fields to guide corrections in areas with minimal clear-sky information.

**Response:** Thank you for your constructive comments. In response to your concern, we provide our reply from two perspectives.

**(1) Regarding the concern that using Himawari-8/9 (H8/9) clear-sky TPW as a reference might introduce additional systematic errors into the fused product:**

H8/9 was selected as the reference field mainly because of its high accuracy, strong temporal continuity, high spatial resolution, and stable retrieval performance. In the revised manuscript (Lines 501–502), we added Table 2, which compares the validation results of H8 TPW, hourly-resampled multi-source microwave TPW (MW TPW), and the fused TPW against GNSS TPW data for 2017.

Table 2. Comparison of hourly TPW from H8 TPW, MW TPW, and fused TPW against GNSS observations in 2017.

| Weather    | Data true | D    | Bias (mm) | RMSE | RRMSE | N        |  |
|------------|-----------|------|-----------|------|-------|----------|--|
| conditions | Data type | R    |           | (mm) | (%)   | N        |  |
| Clear sky  | H8 TPW    | 0.95 | 0.38      | 1.94 | 27.24 | 143670   |  |
|            | MW TPW    | 0.89 | -1.82     | 3.79 | 54.37 | 42367    |  |
| Cloudy     | MW TPW    | 0.88 | -4.31     | 6.44 | 55.44 | 46545    |  |
|            | Fused     | 0.05 | -2.78     | 4.92 | 42.32 | 1.420.62 |  |
|            | TPW       | 0.95 |           |      |       | 142063   |  |

As shown in Tab. 2, under clear-sky conditions, MW TPW has an RMSE of 3.79 mm, while H8 TPW shows a much smaller value of 1.94 mm. Under cloudy conditions, MW TPW has an RMSE of 6.44 mm, whereas the fused TPW improves to 4.92 mm, representing a 23.6 % reduction in error. This demonstrates that using the high-accuracy H8 TPW effectively reduces uncertainty in the fused data under cloudy conditions.

In addition, the multi-source microwave observations have a native temporal sampling of 1–3 hours and a coarse spatial resolution of 0.25°, while H8/9 provides continuous hourly observations with a 2 km resolution—the only dataset covering the entire Tibetan Plateau at such scales. Therefore, H8 supplies high-resolution, stable water-vapor information that the microwave data alone cannot provide. In the fusion framework, microwave data serve as a complementary source to fill the cloudy gaps of H8/9, whereas H8/9 acts as the primary reference for ensuring spatial and temporal consistency. Hence, H8/9 is not an additional error source but rather the most reliable and stable foundation for constraining multi-sensor consistency and enhancing the overall resolution of the fused TPW product.

**(2) Regarding the performance of the adaptive correction under persistent cloud cover:**

We agree with the reviewer that in regions with long-lasting and extensive cloud systems, the adaptive correction may become less effective due to the lack of sufficient clear-sky pixels, which can increase local uncertainty. Future improvements will focus on incorporating cloud physical parameters and short-term numerical weather prediction fields to further enhance correction accuracy under such conditions.

To clarify this point, we have added a new paragraph in the Discussion section (Lines 670–694) describing the current applicability of the algorithm and its potential extensions, as

follows:

**"5.2 Limitations and future improvements of the fusion algorithm**

Although the proposed fusion algorithm successfully reconstructs continuous all-weather TPW fields with high temporal and spatial resolution over the Tibetan Plateau, several limitations remain to be addressed.

First, in the western part of the Tibetan Plateau, the original satellite observations show extensive data gaps because of the limited geostationary coverage and the sparse overpasses of microwave sensors. As a result, the available information for spatiotemporal interpolation is relatively insufficient, which may lead to reduced reconstruction accuracy in these regions. Future work will incorporate additional satellite observations—such as data from Fengyun-4A/B (FY-4A/B) and other geostationary missions—to improve the spatial and temporal coverage of the input datasets, thereby enhancing the quality of reconstructed TPW fields in regions with limited observations.

Second, in regions with persistent and extensive cloud cover, the adaptive correction may experience reduced effectiveness due to the scarcity of valid clear-sky reference pixels from Himawari-8/9 (H8/9). This can lead to locally increased uncertainties, particularly in areas with frequent deep convective systems such as the southern Plateau. Future improvements will focus on introducing additional physical constraints, including cloud microphysical parameters retrieved from infrared or microwave cloud products, and assimilating short-term numerical weather prediction (NWP) fields to enhance correction robustness and continuity under prolonged cloudy conditions.

Third, the current algorithm has been optimized for the Tibetan Plateau, focusing on high-resolution water vapor reconstruction to support regional atmospheric and hydrological studies. Future development can extend this framework toward a near-global scale by integrating multi-geostationary satellite observations to achieve hourly TPW coverage across most low- and mid-latitude regions. For high-latitude areas where geostationary satellites lack coverage, clear-sky TPW retrievals from polar-orbiting optical sensors such as MODIS and MERSI can be incorporated as complementary sources.

In the long term, developing a globally consistent, high-resolution, and purely satellite-based TPW fusion framework will establish a solid observational foundation for quantitative studies of atmospheric moisture transport, energy balance, and land-atmosphere coupling, and will further support the refinement of precipitation (Cui et al., 2025; Ji et al., 2025a), cloud property (Tana et al., 2023, 2025), and radiation estimation (Letu et al., 2023) algorithms across multiple spatial and temporal scales."

**Comment 2:** It is not clear why PWV data from polar imagers such as MODIS or MERSI are not included in the analysis. These valuable data should provide a critical reference or add new information to the fused product.

Response: Thank you for your constructive comments. MODIS and MERSI provide two types of TPW products: thermal infrared (TIR) and near-infrared (NIR). Both are limited to clear-sky conditions and have significant constraints over the Tibetan Plateau. Specifically, taking MODIS as an example, it provides only two effective overpasses per day (one daytime and one nighttime). The TIR product can be retrieved both day and night but has a coarse spatial resolution (~5 km) and lower accuracy over complex terrain. The NIR product offers higher spatial resolution (~1 km) but is available only under daytime clear-sky conditions and is strongly affected by high surface albedo and frequent cloud cover over the Plateau, resulting in limited valid samples.

In contrast, the H8/9 TPW product provides continuous full-disk coverage every 10 minutes with a spatial resolution of about 2 km, offering stable, high-frequency, and high-accuracy water-vapor observations that serve as a reliable reference for multi-source fusion.

We conducted an hourly validation using GNSS TPW data in 2017 to compare the accuracy of the H8/9, MODIS-NIR, and MODIS-TIR TPW products. The results are summarized below:

Table. Validation of MODIS-NIR, MODIS-TIR, and Himawari-8 TPW against GNSS observations in 2017.

| Product | R    | Bias (mm) | RMSE (mm) | RRMSE (%) | N    |
|---------|------|-----------|-----------|-----------|------|
| NIR     | 0.91 | 0.86      | 3.05      | 43.57     | 2508 |

| TIR | 0.80 | -2.35 | 4.61 | 63.11 | 6501   |
|-----|------|-------|------|-------|--------|
| Н8  | 0.95 | 0.38  | 1.94 | 27.24 | 143670 |

The time-matching window was set to ±15 minutes, and the spatial matching radius to 5 km. Among the three datasets, H8/9 shows the highest correlation with GNSS observations at 0.95, followed by MODIS NIR at 0.91 and MODIS TIR at 0.80. For bias, H8/9 exhibits the smallest deviation of 0.38 mm, MODIS NIR shows a slightly larger positive bias of 0.86 mm, and MODIS TIR presents a clear negative bias of -2.35 mm. In terms of overall accuracy, H8/9 again performs best with an RMSE of 1.94 mm and an RRMSE of 27.24%, whereas MODIS NIR and MODIS TIR show larger errors of 3.05 mm (43.57%) and 4.61 mm (63.11%), respectively. Considering that MODIS and MERSI neither provide additional temporal observations nor achieve higher validation accuracy compared with H8/9, incorporating these datasets into the fusion framework may not produce a positive influence on the final results. Therefore, these datasets were not incorporated into the current fusion framework.

Nevertheless, MODIS offers valuable high-resolution NIR observations over polar regions where H8/9 coverage is unavailable. As noted in the revised manuscript, future work will explore integrating MODIS or MERSI TPW observations into a global fusion framework. To address this point, the following text has been added to the Discussion section (Lines 685–689) of the revised manuscript:

"Third, the current algorithm has been optimized for the Tibetan Plateau, focusing on high-resolution water vapor reconstruction to support regional atmospheric and hydrological studies. Future development can extend this framework toward a near-global scale by integrating multi-geostationary satellite observations to achieve hourly TPW coverage across most low- and mid-latitude regions. For high-latitude areas where geostationary satellites lack coverage, clear-sky TPW retrievals from polar-orbiting optical sensors such as MODIS and MERSI can be incorporated as complementary sources."

**Comment 3:** Regarding validation, while the use of 44 GNSS stations is valuable, their sparse and uneven distribution, particularly over western and northern TP, limits the ability to

comprehensively assess the product's accuracy across all topographic and meteorological regimes. The validation might not fully capture errors in the most data-scarce regions. Supplementing the evaluation with data from intensive field campaigns (for example, Scientific Expedition on the Tibetan Plateau), additional independent satellite retrievals, or a cross-validation study during periods with varied cloud cover would strengthen the confidence in the product's performance.

**Response:** Thank you for this valuable suggestion. In the revised manuscript, we have supplemented the validation by adding three new observational datasets — Integrated Global Radiosonde Archive (IGRA) TPW, and two types of TPW data from scientific expeditions over the Tibetan Plateau: (1) microwave radiometer (MWR) TPW over the Tibetan Plateau, and (2) GNSS TPW from the central Himalayas — and included their corresponding validation analyses.

To clearly describe these datasets, two new subsections have been added to the Data section of the revised manuscript (Lines 269–283), as follows:

**"2.2.4 IGRA radiosonde TPW data**

The Integrated Global Radiosonde Archive (IGRA) provides vertical profiles of temperature, humidity, and pressure at multiple levels. In this study, the TPW derived from IGRA observations in 2022 was used to independently validate the fused TPW dataset. The IGRA data were obtained from the National Climatic Data Center (Durre et al., 2006).

**2.2.5 Scientific expedition ground-based TPW observations**

Two types of ground-based observations from the scientific expedition were employed to further evaluate the fused TPW data:

- (1) Microwave radiometer (MWR) observations: Continuous tropospheric observations (0–10 km) were obtained from the MWR network deployed across the Plateau during 2021–2022 (Chen & Ma, 2022; Chen et al., 2024). The instruments provide profiles of temperature and humidity at 58- or 83-layer vertical resolutions, from which TPW is derived. Observation sites include MAWORS, NADOR, Mangai, Naqu, Changdu, Leshan, QOMS, SETS, and Kabu.
- (2) GNSS TPW observations in the central Himalayas: A five-station north-south GNSS chain was established along the Yadong-Lhasa fault zone (XYDX, DNLG, SMDX, JZLZ,

and LKZZ), providing high-accuracy (±0.1 mm) TPW retrievals from 2015 to 2019 (Wang et al., 2019; Yang, 2023). These data were used to validate the fused TPW performance in steep terrain regions of the central Himalayas."

All three datasets were used for independent validation of the fused TPW product at the hourly scale.

Furthermore, the validation results from these newly added datasets were incorporated into a new subsection "4.1.3 Additional validation using radiosonde and scientific expedition data" (Lines 539–566) of the revised manuscript. The added text reads as follows:

[revised manuscript text omitted]

---

## Referee Report (RR1)

Comments to the Author,

The authors addressed most of my questions and concerns. The manuscript is greatly improved. However, it is still one suggestion that needs the author's attention before accepting the manuscript.

The authors use the Himawari-8/9 (H8/9) clear-sky TPW product as the foundational reference for both bias correction and spatial downscaling, from which the fused TPW data under cloudy conditions are subsequently derived. Therefore, it is recommended that the authors provide a separate evaluation of the fused TPW specifically under clear-sky and cloudy conditions to more comprehensively assess its accuracy and reliability.

In the current version of the manuscript, although the authors evaluate the fused TPW data under all-weather conditions for the period 2016–2022, the separate evaluations for clear-sky and cloudy conditions are only provided for the year 2017. It is recommended that the authors extend this separate evaluation to the same years (2016–2022) to ensure the robustness and consistency of the dataset.

---

## Author Response (AR2)

Dear Editors and Reviewers,

We sincerely thank the editors and the anonymous reviewers for their constructive and insightful comments on our manuscript entitled "*An hourly 0.02° total precipitable water dataset for all-weather conditions over the Tibetan Plateau through the fusion of observations of geostationary and multi-source microwave satellites*" (Manuscript ID: essd-2025-365).

We carefully considered the reviewer's remaining suggestion, which recommended extending the separate evaluations of the fused TPW dataset under clear-sky and cloudy-sky conditions to cover all years from 2016 to 2022. We fully agreed with this valuable advice and have accordingly revised the manuscript by adding new statistical validation results and corresponding analyses. All modifications have been incorporated into the revised manuscript, with changes clearly marked for your review.

We believe these revisions have further improved the quality and completeness of the manuscript. Thank you for your time and effort in handling our submission, and we look forward to your further evaluation.

Thank you and best regards,
Dabin Ji
E-mail: jidb@aircas.ac.cn
Affiliation: State Key Laboratory of Remote Sensing and Digital Earth,
Aerospace Information Research Institute, Chinese Academy of Sciences,
Beijing 100101, China

**Response to Reviewer Comments**

The authors addressed most of my questions and concerns. The manuscript is greatly improved. However, it is still one suggestion that needs the author's attention before accepting the manuscript.

**Response:** We sincerely thank the reviewer for providing these additional constructive comments and for the positive assessment of our previous revisions.

**Comment 1:** The authors use the Himawari-8/9 (H8/9) clear-sky TPW product as the foundational reference for both bias correction and spatial downscaling, from which the fused TPW data under cloudy conditions are subsequently derived. Therefore, it is recommended that the authors provide a separate evaluation of the fused TPW specifically under clear-sky and cloudy conditions to more comprehensively assess its accuracy and reliability.

In the current version of the manuscript, although the authors evaluate the fused TPW data under all-weather conditions for the period 2016 – 2022, the separate evaluations for clear-sky and cloudy conditions are only provided for the year 2017. It is recommended that the authors extend this separate evaluation to the same years (2016 – 2022) to ensure the robustness and consistency of the dataset.

**Response:** Thank you for this valuable suggestion. We fully agree that separate and extended evaluation of clear-sky and cloudy-sky conditions is critical to verifying the dataset's robustness. We have revised the manuscript by supplementing statistical validation results of the fused TPW dataset under clear-sky and cloudy-sky conditions for the entire 2016–2022 period, and added corresponding analysis in the results section.

Specifically, two new tables (Table 5 and Table 6) have been added to present the statistical metrics (R, Bias, RMSE, RRMSE, and sample size N) for clear-sky and cloudy-sky conditions across all years. Meanwhile, relevant analysis content has been supplemented in Section 4.1.4 (Validation of the fused TPW dataset for all years, Lines 566–586) to compare the performance of the fused dataset under different weather conditions and confirm its interannual consistency. The revised content is as follows:

"**4.1.4 Validation of the fused TPW dataset for all years**

To comprehensively evaluate the reliability of the fused TPW dataset, statistical validation was performed against GNSS observations from 2016 to 2022 under three weather conditions: all-weather, clear-sky, and cloudy-sky. The results are presented in Tab. 4 (all-weather), Tab. 5 (clear-sky), and Tab. 6 (cloudy-sky).

**Table 4. Statistical validation of the fused TPW dataset against GNSS observations under all-weather conditions for all years (2016–2022).**

| Year | R | Bias (mm) | RMSE (mm) | RRMSE (%) | N |
|------|------|-------|------|-------|--------|
| 2016 | 0.91 | -1.49 | 3.98 | 43.09 | 253194 |
| 2017 | 0.91 | -1.39 | 3.82 | 40.38 | 285733 |
| 2018 | 0.92 | -1.23 | 3.82 | 40.43 | 311143 |
| 2019 | 0.91 | -1.02 | 3.37 | 39.99 | 174298 |
| 2020 | 0.91 | -1.39 | 4.24 | 38.68 | 201789 |
| 2021 | 0.92 | -0.99 | 3.50 | 38.69 | 300686 |
| 2022 | 0.92 | -1.04 | 3.56 | 39.37 | 279514 |

**Table 5. Statistical validation of the fused TPW dataset against GNSS observations under clear-sky conditions for all years (2016–2022).**

| Year | R | Bias (mm) | RMSE (mm) | RRMSE (%) | N |
|------|------|-------|------|-------|--------|
| 2016 | 0.92 | -0.06 | 2.41 | 35.40 | 124831 |
| 2017 | 0.95 | 0.38 | 1.94 | 27.24 | 143670 |
| 2018 | 0.95 | 0.27 | 2.00 | 27.37 | 142859 |
| 2019 | 0.95 | 0.30 | 1.85 | 27.74 | 78360 |
| 2020 | 0.95 | 0.28 | 2.00 | 24.72 | 106901 |
| 2021 | 0.96 | 0.43 | 1.82 | 26.81 | 138246 |
| 2022 | 0.96 | 0.30 | 1.80 | 26.52 | 136784 |

**Table 6. Statistical validation of the fused TPW dataset against GNSS observations under cloudy-sky conditions for all years (2016–2022).**

| Year | R | Bias (mm) | RMSE (mm) | RRMSE (%) | N |
|------|------|-------|------|-------|--------|
| 2016 | 0.91 | -2.74 | 4.95 | 43.51 | 140840 |

| 2017 | 0.95 | -2.78 | 4.92 | 42.32 | 142063 |
|------|------|-------|------|-------|--------|
| 2018 | 0.92 | -2.50 | 4.85 | 43.10 | 168284 |
| 2019 | 0.92 | -2.10 | 4.23 | 42.81 | 95938 |
| 2020 | 0.90 | -3.28 | 5.81 | 40.84 | 94988 |
| 2021 | 0.92 | -2.20 | 4.46 | 40.59 | 162440 |
| 2022 | 0.92 | -2.33 | 4.66 | 41.63 | 142730 |

Under all-weather conditions (Tab. 4), the fused TPW dataset exhibited stable performance throughout the study period, with correlation coefficients remaining within 0.91 – 0.92, RMSE values within 3.37 and 4.24 mm, and slightly dry biases around −0.99 to −1.49 mm. Note that fewer validation samples were available in 2019 and 2020 due to reduced GNSS observations, rather than changes in the fused dataset. Under clear-sky conditions (Tab. 5), the correlation coefficient ranged from 0.92 to 0.96. The Bias was between -0.06 mm and 0.43 mm. The RMSE was from 1.80 mm to 2.41 mm, and the RRMSE ranged from 24.72% to 35.40%. The accuracy under clear-sky conditions is higher than that under all-weather conditions. Under cloudy-sky conditions (Tab. 6), the correlation coefficient ranged from 0.90 to 0.95. The RMSE ranged from 4.23 mm to 5.81 mm, with a relatively higher value in 2020. However, the RRMSE in 2020 was 40.84%, which fell within the fluctuation range of 40.59% to 43.51% observed in other years without abnormal deviation. Overall, the fused TPW dataset showed relatively stable accuracy across different years and weather conditions."

---

## Author Response (AR3)

Dear Editors and Reviewers,

We are extremely delighted to receive the acceptance notification of our manuscript. First and foremost, we would like to express our sincere gratitude to you and the anonymous reviewers for your valuable suggestions and meticulous efforts devoted to our work. These insightful comments have played a crucial role in enhancing the quality of our manuscript.

Prior to the formal publication of the paper, we conducted a thorough recheck of the manuscript and unfortunately identified some writing errors. We sincerely apologize for these oversights. To ensure the accuracy and rigor of the paper, we have revised all these errors carefully, and the specific revisions are detailed as follows (with the modified content marked in red):

1.Line 5: Adjust the position of "and" in the author list Revised author list:
"Qixiang Sun[1,2], Husi Letu[1, *], Yongqian Wang[3], Peng Zhang[4, 5], Hong Liang[4], Chong Shi[1], Shuai Yin[1], Jiancheng Shi[6], and Dabin Ji[1, *]"

2.Line 277: Changed the uppercase "C" in "Continuous" to lowercase "c".

3.Line 279: Removed the redundant "TPW" abbreviation.

4.Adjust incorrect section numbers Corrected the wrong section numbers as follows:
"2.2.4 MIMIC-TPW2 data" → "2.2.6 MIMIC-TPW2 data"
"2.2.5 ERA5 TPW data" → "2.2.7 ERA5 TPW data"
"2.2.6 Auxiliary data" → "2.2.8 Auxiliary data"
"4.2.2 Comparison with other TPW products" → "4.1.2 Comparison with other TPW products"

5.Line 698: Supplement reference "Shi et al. (2025)". This reference was originally a reference for the study but was inadvertently omitted in the previous version.

After completing the revisions, we have re-uploaded the updated manuscript in both PDF and TXT formats, which are clean versions

Once again, we would like to extend our heartfelt thanks to you and the reviewers for your precious suggestions and the significant time you have invested in our paper.

Best regards,
Dabin Ji

E-mail: jidb@aircas.ac.cn

Affiliation: State Key Laboratory of Remote Sensing and Digital Earth,
Aerospace Information Research Institute, Chinese Academy of Sciences,
Beijing 100101, China